# Trans-illumination intestine projection imaging of intestinal motility in mice

Depeng Wang[1,2], Huijuan Zhang[1], Tri Vu[1], Ye Zhan[1], Akash Malhotra[3], Pei Wang[3], Upendra Chitgupi[1], Aliza Rai[3], Sizhe Zhang[1], Lidai Wang [ORCID][3], Jan D. Huizinga [ORCID][4], Jonathan F. Lovell[1] & Jun Xia [ORCID][1✉]

Functional intestinal imaging holds importance for the diagnosis and evaluation of treatment of gastrointestinal diseases. Currently, preclinical imaging of intestinal motility in animal models is performed either invasively with excised intestines or noninvasively under anesthesia, and cannot reveal intestinal dynamics in the awake condition. Capitalizing on near-infrared optics and a high-absorbing contrast agent, we report the Trans-illumination Intestine Projection (TIP) imaging system for free-moving mice. After a complete system evaluation, we performed in vivo studies, and obtained peristalsis and segmentation motor patterns of free-moving mice. We show the in vivo typical segmentation motor pattern, that was previously shown in ex vivo studies to be controlled by intestinal pacemaker cells. We also show the effects of anesthesia on motor patterns, highlighting the possibility to study the role of the extrinsic nervous system in controlling motor patterns, which requires una- nesthetized live animals. Combining with light-field technologies, we further demonstrated 3D imaging of intestine in vivo (3D-TIP). Importantly, the added depth information allows us to extract intestines located away from the abdominal wall, and to quantify intestinal motor patterns along different directions. The TIP system should open up avenues for functional imaging of the GI tract in conscious animals in natural physiological states.

---

[1] Department of Biomedical Engineering, University at Buffalo, State University of New York, Buffalo, NY, USA. [2] Department of Biomedical Engineering, Duke University, Durham, NC, USA. [3] Department of Mechanical and Biomedical Engineering, City University of Hong Kong, Hong Kong, China. [4] Farncombe Family Digestive Health Research Institute, Department of Medicine, McMaster University, Ontario, Canada. ✉email: junxia@buffalo.edu

Gastrointestinal (GI) diseases affect more than 50 million people and account for millions of clinical visits annually in the United States[1,2], resulting in a serious economic impact and significant social cost. Irritable bowel syndrome (IBS) and inflammatory bowel disease are two GI diseases that cause motility dysfunction in the intestines. Diarrhea-predominant IBS (IBS-D) and constipation-predominant IBS (IBS-C) are associated with abnormal motility that has an imbalance between propulsion and segmentation. Motor dysfunction is found in chronic refractory constipation and intestinal pseudo-obstruction. Because abnormal motility is linked to severe symptoms and is the primary target for therapy, motility has been investigated extensively[3,4], but mostly invasively in preclinical studies in animal models. In particular, mouse models are widely used for intestinal imaging studies, because many established disease models have been developed and many knock-out and knock-in models of genetic subtypes are available[5]. Many studies also used mice to uncover the mechanims underlying intestinal motility involving the nervous systems and intestinal pacemaker activities[6–8].

Invasive methods, such as in vitro imaging modalities of intestine[9,10], have been extensively used. While they have played an important role in pharmacological experiments on intestinal motility, the terminal nature of these imaging methods prevents the longitudinal assessment of disease progression and therapeutic responses. It is uncertain that the motor patterns observed in vitro identically represent the patterns that happen in vivo since all in vitro studies require the intestines to be disconnected from the central nervous system, which plays a crucial role in all motor activities. Hence, critically important investigations of brain-gut-communication require free-moving unanesthetized animal studies. In vivo techniques using X-ray[11] or magnetic resonance imaging (MRI)[12] are also problematic because of the need for anesthesia or movement restraint.

This study introduces Trans-illumination Intestine Projection (TIP) imaging, which we have developed the intestinal imaging system for free-moving mice (2D-TIP, Fig. 1a; left and Supplementary Fig. 1) and the 3D light-field-based intestinal imaging system (3D-TIP, Fig. 1a; right). Capitalizing on near-infrared (NIR) optics and a novel contrast agent, TIP identified features located in deep tissue with high imaging contrast. Specifically, 2D-TIP enables the visualization of peristalsis and segmentation motor patterns of free-moving mice, which has never been reported before. This unique imaging ability of 2D-TIP allows the in vivo demonstration that the anesthetized mice exhibited a much slower intestinal motility rate than the awake mice, highlighting its ability to study central nervous system regulation. Equipped with the emerging light-field technique, 3D-TIP can visualize the volumetric distribution of intestine in one snapshot. The volume imaging capability of 3D-TIP permits the extraction of intestines located far away from the abdomen wall, and allows the quantification of intestinal motility along different directions. We anticipate that TIP method will open up more avenues for functional imaging of the GI tract in animal models.

## Results

**TIP imaging system.** Figure 1a shows a schematic of the TIP imaging system. During free-moving mouse imaging, the mouse moves on the imaging platform. A tracking camera identifies the position of the free-moving mouse by detecting two painted markers on the back of the mouse (Fig. 1b). The tracking camera then sends the mouse's position to a Raspberry PI unit, which controls an $x$–$y$ motor system to move the laser spots to the intestine region. To precisely track the movement of the mouse and control light illumination, all components operate in a closed-loop as shown in Fig. 1d. The NIR camera detects transillumination images through a set of filters and polarizers (Fig. 1c). The imaging contrast originates from a highly optical absorbing material [butyl-2,3-naphthalocyanine (BNc) micelles][3], which was gavaged into the mouse. Due to weak absorption of biological tissue in the NIR region, the 808 nm light penetrates through the intestinal region of the mouse, where no endogenous absorption at that wavelength exists, resulting in a bright area in the NIR image. As for intestines filled with BNc, the 808 nm light will be absorbed, resulting in a shadow in the NIR image (Fig. 1c). To ensure that the intestine is filled with contrast, our experiment followed the timeline shown in Fig. 1e.

To verify the imaging resolution of TIP in scattering-free medium, we imaged a United States Air Force (USAF) resolution target in air and quantified a resolution of 99.2 µm (Supplementary Fig. 2). To verify the imaging depth of TIP, we imaged BNc-filled tubes embedded in agar gels, which mimic both absorption and scattering of biological tissue (Supplementary Figs. 3 and 4 and "Methods"). The results indicate that TIP can visualize the 0.5-mm-inner-diameter tube at up to 8 mm depth. This imaging depth covers the majority of intestines underneath the abdomen wall[13]. The mouse intestine has an average diameter of 4 mm[14] in the resting condition, but the diameter dramatically decreases during contraction. We experimentally validated this by quantifying the diameter of a contracted intestine which is 25% of the 4 mm averaged intestine diameter (Supplementary Fig. 5). This observation suggested a 75% change (3 mm) in intestine diameter, which is large enough for TIP to capture.

**2D-TIP imaging of free-moving mouse.** 2D-TIP was further validated through in vivo imaging of mice at 21 min, 70 min, and 2.5 h post gavage of contrast agent (Fig. 2a). The data preprocessing procedures are listed in the "Methods" section and Supplementary Figs. 6 and 7. Twenty-one minutes post gavage, 2D-TIP showed that the intestine started to be filled with contrast (Fig. 2a: top). At this stage, the dominant intestinal motility is peristalsis-driven slow wave, which originates from the pacemaker cells and the interstitial cells of Cajal[15,16]. The peristaltic activity is a wavelike movement that pushes the contrast agent forward and is shown as propagating bands in the spatial-temporal map (Fig. 2b: top)[17,18].

Thirty minutes post gavage, 2D-TIP showed that more sections of intestine became filled with contrast (Supplementary Fig. 8), consistent with previous study[19]. Seventy minutes post-administration (Fig. 2a: middle), more regions of the intestine showed up and motor patterns indicated a mixture of peristalsis and segmentation (Fig. 2b: middle). Two hours and thirty minutes post gavage (Fig. 2a: bottom), the spatial-temporal map showed the classic Cannon-type segmentation pattern (Fig. 2b: bottom, and Supplementary Fig. 9), which is similar to that obtained in ex vivo experiments[20]. For comparison, a mouse was also imaged under anesthesia (Fig. 2c). The segmentation pattern of the anesthetized mouse was similar to that of the awake mouse, but the contraction frequency was much lower (Fig. 2d and Supplementary Fig. 10). To better reveal the difference, we extracted the average pixel value of one intestinal cross-section for both the anesthetized and free-moving mice (2.5 h post gavage). Within a 15 s time window, the anesthetized mouse exhibited a significantly slower contraction rate when compared to that of the free-moving mouse (Fig. 2e). The frequency analysis of segmentation rates is shown in Fig. 2f and Supplementary Fig. 11. The peak frequency for the anesthetized mouse was 0.380 ± 0.085 Hz, which agrees with values reported in other studies[21]. For the free-moving mice, the peak frequency was 0.772 ± 0.089 Hz, which was two times the rate of anesthetized mice.

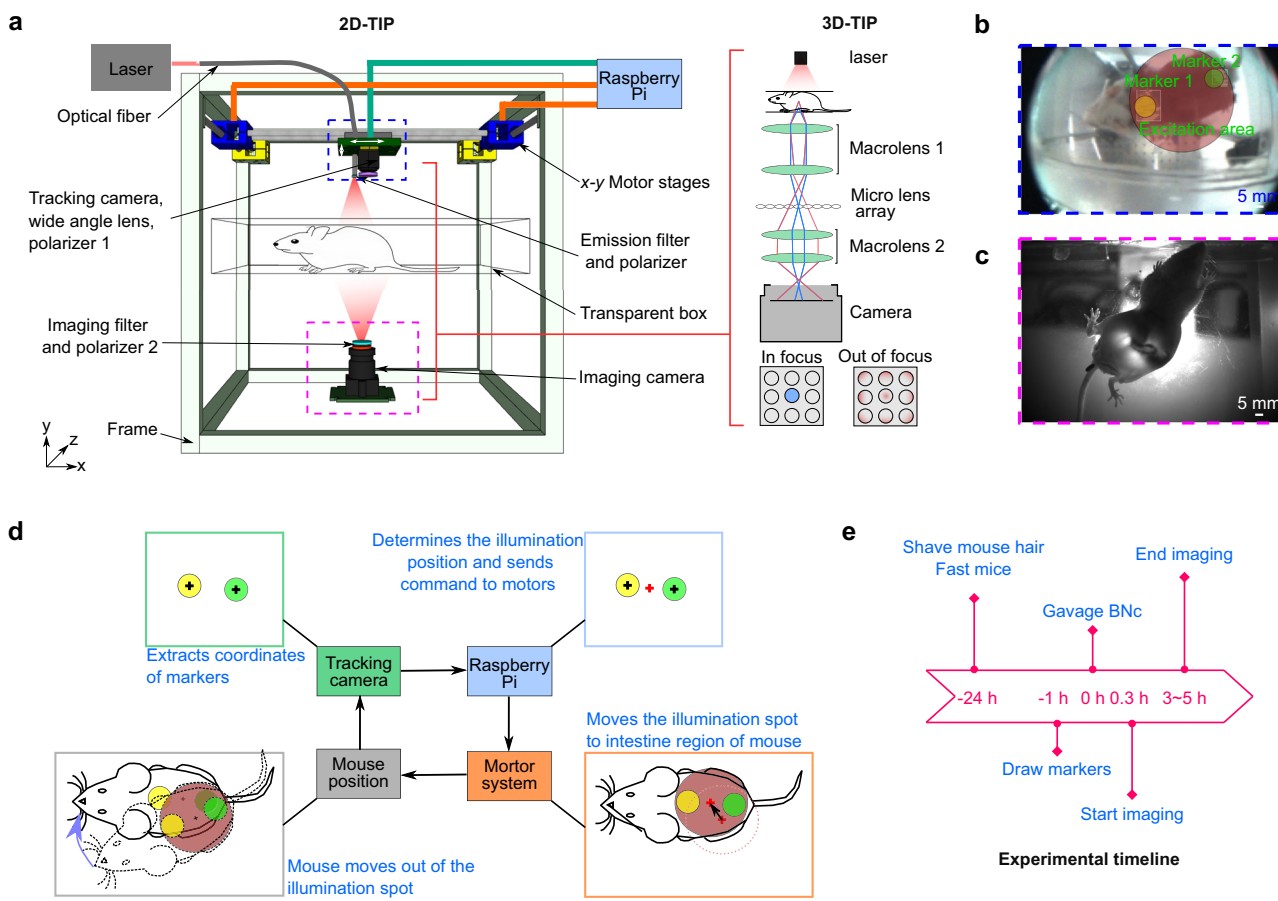

**Fig. 1 TIP imaging system. a** A schematic drawing of the 2D-TIP system that is capable of imaging free-moving mice. The inset on the right illustrates the schematic of 3D-TIP, which is capable of imaging volumetric intestine. The microlens array captures 4D light-field information. When the point source is located at the principal focal plane (blue), it will be imaged by only one microlens (bottom left); when the point is located out of the principal focal plane (red), it will be imaged by multiple microlenses (bottom right). The light-field information allows for 3D image reconstruction. **b** An exemplary image acquired by the tracking camera in 2D-TIP. **c** An exemplary image acquired by the bottom NIR camera. **d** A diagram illustrating the closed-loop control of the tracking system. **e** The experimental timeline.

To better reveal the movement of the intestine, we show its activity at different times, which clearly represents the movement of contrast agent in the intestine. Within a time window of 120 s, we observed pacemaker-driven peristalsis, which causes propulsion to move content towards the anal direction[20]. Due to peristalsis, the contrast agent moved rapidly over a distance of more than 20 mm, filling most sections of the intestine shown in Fig. 3a. Within a shorter time window (1.2 s), we observed a detailed process of segmentation (Fig. 3b). Due to simultaneous transient contractions in the left and right of the orange circled intestine regions ($t = 0$ s), contrast agents are pushed to move towards each other. Then, a contraction appears in between the original contractions (Fig. 3b, $t = 0.67$ s) and segments the contrast agent, completing one cycle of segmentation.

**Assessment of TIP for long-duration imaging.** Long-duration imaging is important for continuous monitoring of intestinal motility and changing motor patterns, however, there are challenges. A potential concern is that long-time illumination of a mouse with the light intensity required by TIP might cause a temperature rise in the mouse body and affect mouse behavior, which could prevent long-time recording. Another challenge is that the distribution of the intestine inside the abdominal cavity will change while the mouse is moving freely. These changes might affect the extraction of the intestine for spatial-temporal map calculation. To assess whether TIP can overcome these challenges, we performed two tests. In our first test, we illuminated a piece of chicken breast tissue for 2 h at a laser intensity of 14 mW/cm² (the same intensity used in the imaging experiment) and continuously monitored the surface temperature using a thermal camera (FLIR one). As a control, we also imaged another piece of chicken breast tissue without laser illumination. The result indicated that there was no temperature rise in the exposed tissue and no difference between the exposed and control tissues (Supplementary Fig. 12), proving that long-time illumination is not a concern for TIP, at least from a photothermal hyperthermia perspective. Our second test was to extract the intestine profiles from TIP images involving different intestine distributions (Fig. 4a). Although the entire intestine moves in response to different animal behaviors, TIP could clearly capture the intestine due to the high absorbance of the contrast agent. To digitally extract the intestine, we first calculated the similarity of all frames to separate different behaviors (Fig. 4b, Supplementary Fig. 13a). We then extracted the intestine under each behavior and computed the spatial-temporal map. We combined all spatial-temporal maps to form a long-time map (Fig. 4c and Supplementary Fig. 13b), which proves that TIP is capable of long-duration imaging. The calculation of long-duration motor pattern for anesthetized mice is easier and does not require tracking of the moving intestine (Fig. 4d and Supplementary Fig. 13c).

Once we obtained the motor patterns, we compared the intestinal motility between free-moving mice and anesthetized

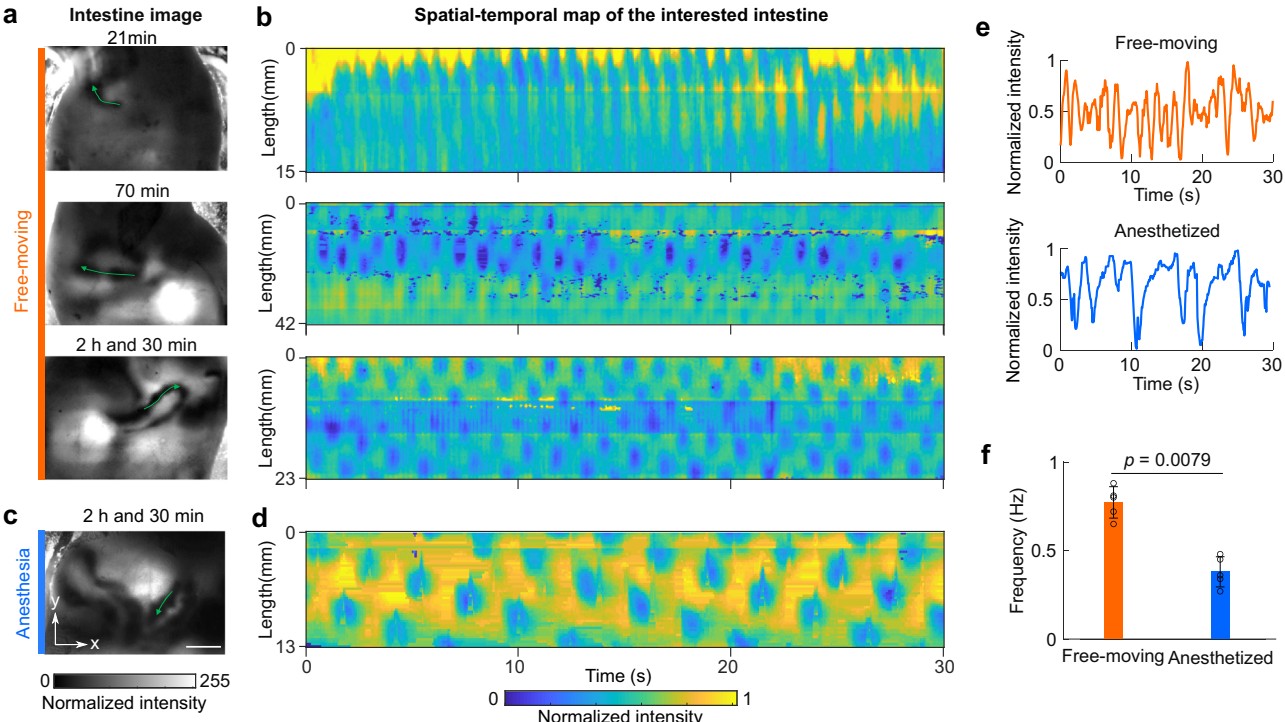

**Fig. 2 2D-TIP imaging of intestinal motility in free-moving and anesthetized mice. a** Temporal intestinal images of a free-moving mouse. **b** Spatial-temporal maps of the intestinal region labeled with green arrows in (**a**), (Supplementary Videos 1, 2, and 3). **c** Intestinal images of an anesthetized mouse (green arrow shows the intestinal region of interest). **d** Spatial-temporal map of the interested intestine in (**c**) (Supplementary Video 4). Scale bar: 10 mm. **e** The average intensity of the intestinal cross-sectional profile shows segmentation frequencies of free-moving and anesthetized mice. **f** Frequency analysis reveals that the intestinal segmentation in free-moving mice occurred at a faster rate than that of the anesthetized mice (mean ± s.d., $n = 5$ mice, two-sided Wilcoxon rank-sum test).

mice. For better comparison, we also projected intestines imaged at different time points together to form a map of all the visualized intestines over a time window of 5 h. We identified intestines using the stomach and cecum as landmarks—the duodenum is closer to the stomach, the ileum is connected to the cecum, and the intestine in the middle is jejunum (Fig. 5a). For either the anesthetized or the free-moving mice, we compared the dominant contraction frequency for the same intestine section (Fig. 5b). The duodenum appeared ~30 min post gavage for both the free-moving mice and the anesthetized mice, and the jejunum showed up 1.5 h post gavage. We continued imaging the mice up to 5 h post gavage. Over the imaged time window, both the free-moving mice and the anesthetized mice showed a gradually decreased motility frequency over time, as the contrast moved from duodenum to ileum. This exhibited the intrinsic frequency gradient of the pacemaker activity, which is essential for anal propagation. Compared to the free-moving mice, the anesthetized mice exhibited a lower motility frequency for the same intestinal section (Fig. 5c)[20]. For both the free-moving mice and the anesthetized mice, TIP visualized the intestine filled by the contrast agent at different time points (Fig. 5d, e), providing a panoramic view of the intestine.

**Dual-contrast imaging with TIP**. TIP is also capable of multi-color imaging, by using contrast agents that are spectrally separated in absorbance wavelengths. With the matched illumination wavelength, separate contrast agent can be imaged. This would not be straightforward or even possible using other modalities such as X-ray or MRI. A major advantage of multicolor TIP is the accurate localization of contrast agents at different sections of intestine, thereby eliminating the problem of overlapping sections

of the intestine. As an example, we performed dual-color TIP imaging (setup as described in the "Methods"). We sequentially gavaged two contrast agents with a 30-min interval (Fig. 6a), and imaged the mouse with two wavelengths, semi-simultaneously (Fig. 6b). Within 2.5 h, each contrast agent revealed different intestine sections which clearly represented the upper and the downstream sections of the intestine (Fig. 6c). As the contrast agents propagated inside the intestine, they eventually reached the same section of intestine and mixed with each other at the 5 h time point (Fig. 6c). The system can be further modified to enable three-color or four-color imaging, which will offer more possibilities in studying intestine movement. This solves a major problem with MRI or ultrasound imaging that suffers from the inability to visualize or measure motor patterns from overlapping segments of the intestine or colon.

**TIP resolved overlapped intestines**. The small intestine of the mouse always has overlapping segments that prevent proper in vivo studies of propulsive movements using current techniques. We demonstrate here that TIP resolved overlapped intestines in single-color and dual-color imaging. We classified the overlap of intestine into two cases. One case showed "low overlap" where the two sections of intestine formed a cross with one on top of the other. In this case, we show that the two intestines can be visualized with our TIP (Fig. 7a). The other case showed "high overlap" where one intestine was right above the other and was fully covered during imaging. In this case, our dual-color TIP imaging easily differentiated the highly overlapped two sections of intestines (Fig. 7b). Overlapping intestinal segments are one major reason why in vivo studies have not been widely explored, but this problem is now solved by our TIP.

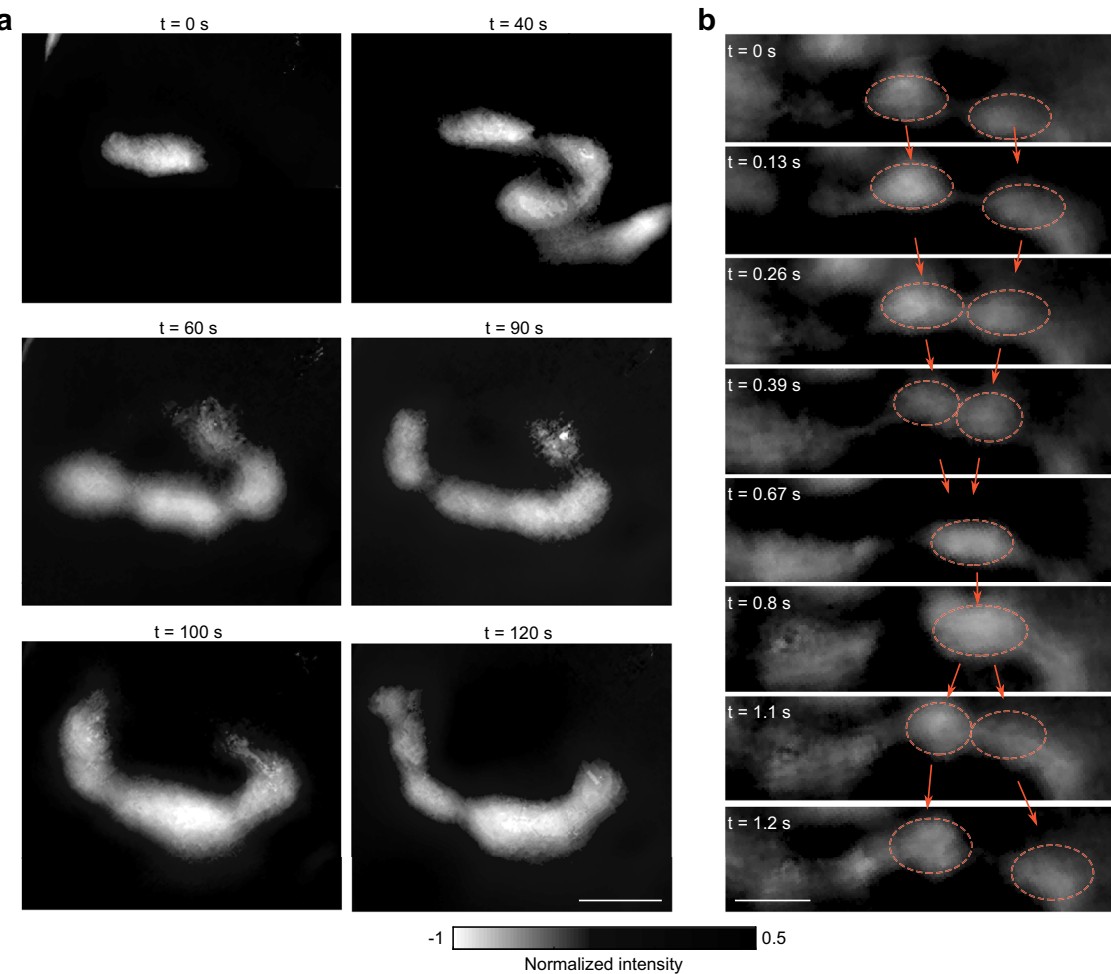

**Fig. 3 Detailed movement of the intestine revealed by 2D-TIP, showing typical pacemaker-driven peristalsis and segmentation. a** Representative 2D-TIP frames within a 120 s time window demonstrating the propagation of contrast agent inside the intestine, driven by a peristaltic motor pattern. **b** 2D-TIP frames showing a typical intestinal segmentation motor pattern. Scale bar: 5 mm.

**3D-TIP imaging of intestine**. The 3D-TIP system uses the same imaging camera and excitation light as the 2D system, but utilizes a different imaging lens set (Fig. 1a: right) for light-field detection. There is a trade-off between the field of view (FOV) and resolution. To achieve a reasonable axial resolution for intestine imaging, the lateral FOV of 3D-TIP was limited to $33 \times 33$ mm$^2$. Therefore, as a proof of concept, we demonstrated 3D-TIP with anesthetized mice. A larger microlens array and a larger camera sensor will enable imaging of free-moving mice in the future. We first validated the axial and lateral resolution of the system by imaging black hair of human. At 3-mm depth, the lateral resolution was 0.5 mm and the axial resolution was 4 mm (Supplementary Fig. 14). The axial resolution at this depth was comparable to the 4 mm diameter of mice intestine[14]. The spatial resolution varied at different depths and the axial resolution degraded to 6 mm at 5-mm depth (Supplementary Fig. 14). To test the 3D imaging performance of the system, we imaged three hairs located at depths of $-2$, 1, and 5 mm, respectively. All three hairs were clearly resolved by 3D-TIP (Fig. 8a).

We then tested 3D-TIP through volumetric intestine imaging in vivo 1 h post gavage. Owing to the depth information provided by light-field imaging, 3D-TIP allowed us to better extract intestines located far away from the abdomen wall (Fig. 8b). For instance, the depth index image clearly identified one section of intestine (labeled with the red box) that was not visible neither in the 0 mm depth image nor in the minimal amplitude projection

(MAP) image. By stacking 2D images over different depths, we obtained a 3D image of two intestines (Fig. 8c), and quantified the spatial-temporal map along both the lateral and axial directions (Fig. 8d–g and Supplementary Fig. 15). We found that the same intestine produced different spatial-temporal maps along these two directions, as the cross-section of the intestine is not perfectly circular inside the body and the contraction is not happening evenly along different directions[22]. Supplementary Video 5 shows the dynamic changes of two intestines in 3D. We also showed the reconstructed intestine image at depths of 4, 0, and $-4$ mm (Fig. 8h), and the spatial-temporal map in Fig. 8i clearly captures the segmentation pattern. The segmentation pattern shows that the short segment activity (blue) propagates relatively slowly (compared to the peristaltic activity). We have previously provided evidence for the hypothesis that the segmentation pattern is generated by the ICC slow waves in the myenteric plexus that interact with a stimulus-induced, much slower propagating slow wave, to generate the segmentation pattern[20,23].

To image movements along the entire length of the intestine, we imaged the mice over 5 h and acquired data at different time points post gavage of contrast agent (Supplementary Fig. 16). Similar to 2D imaging, 3D-TIP visualized sections of intestine over time. To display the depth information, we overlaid the depth index of intestine on top of the intestine image at the principal focal plane (Supplementary Fig. 16). Similar to 2D

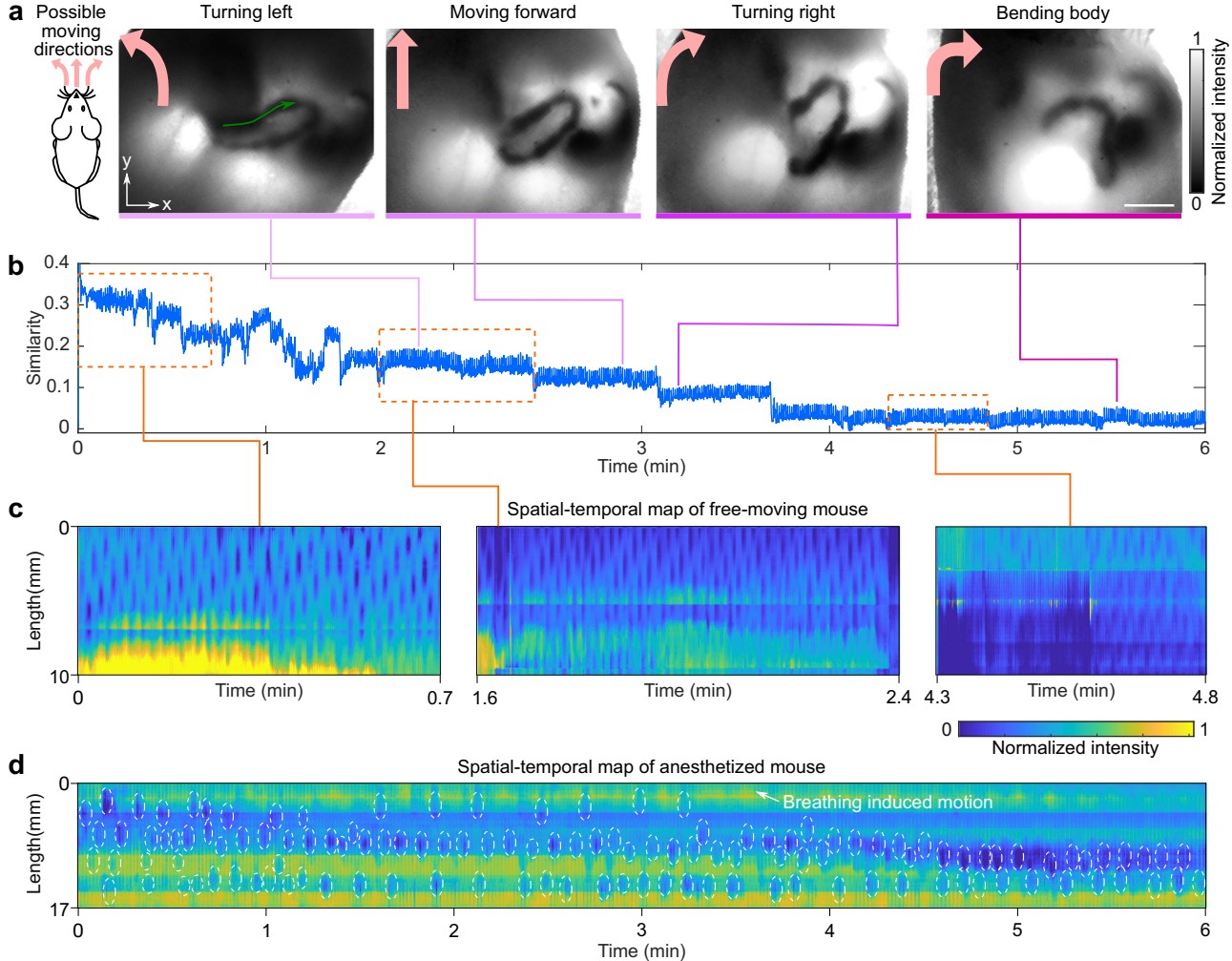

**Fig. 4 TIP demonstrated long-duration imaging of mouse intestine. a** TIP captured the profile of the mouse intestine when the mouse performed different behaviors in a free-moving state. **b** The similarity of frames acquired from the free-moving mouse. The similarity changed when the mouse changed its behavior. **c** Similarity-based data processing generated the long-duration spatial-temporal map for the free-moving mouse. To show the details of the subpattern, only three sections of the whole pattern are shown here. A combined full pattern can be seen in Supplementary Fig. 13a. **d** The spatial-temporal map of the intestine of an anesthetized mouse, acquired over 6 min. White circles indicate intestinal contractions.

imaging, we observed more and more intestine sections as the contrast agent moved inside the intestine.

## Discussion

In this study, we developed a TIP system for intestinal 2D imaging of free-moving mice and 3D imaging of anesthetized mice. Capitalizing on novel NIR contrast agents and optics, our system offers high-contrast and deep-tissue imaging. Our tracking device automatically tracks the mouse movement while the registration algorithm facilitates the data analysis of a moving animal. At different time points post gavage, spatial-temporal maps of the intestine clearly showed the transition of motor patterns over time after filling of the stomach, from peristalsis in the proximal intestine followed by segmentation in the remainder of the intestine to facilitate absorption. We showed that the first motor pattern after gavage is the slow-wave-driven peristalsis. Our experiments represent the demonstration of this phenomenon in freely moving mice and it is similar to that observed in restrained mice using X-rays[18]. Slow-wave-driven peristalsis in the small intestine is orchestrated by pacemaker cells, the interstitial cells of Cajal (ICC) associated with the myenteric plexus[15,16]. Most studies focus on neurally driven motor patterns derived from in vitro

studies[24], but here we show that in free-moving mice peristalsis in the proximal intestine occurs at the frequency of ICC pacemaker activity as proven previously[18]. This is followed by segmentation motor patterns, a very characteristic motor pattern shown to be controlled by two pacemakers that interact with each other[20,23]. In the numerous studies on intestinal motor activity, ex vivo, this motor pattern is almost never observed, a testament to the essential nature of developing methodology with unrestrained mice. We have also demonstrated that an awake mouse exhibits intestinal motility at a higher frequency compared to an anesthetized mouse, highlighting the possibility to study control mechanisms involving the extrinsic nervous system. This study represents the first intestinal imaging in free-moving mice. Future studies will involve the study of motor patterns in the presence of natural content that will induce various neurally driven peristaltic motor patterns[24,25].

Through dual-color TIP imaging, we also demonstrate the feasibility of TIP in differentiating different sections of the intestine. With 3D-TIP, we visualized the volumetric intestine profile and identified an intestine that could be barely seen in 2D imaging, and demonstrated the 3D optical intestine imaging in anesthetized mice. Again, we were able to visualize the

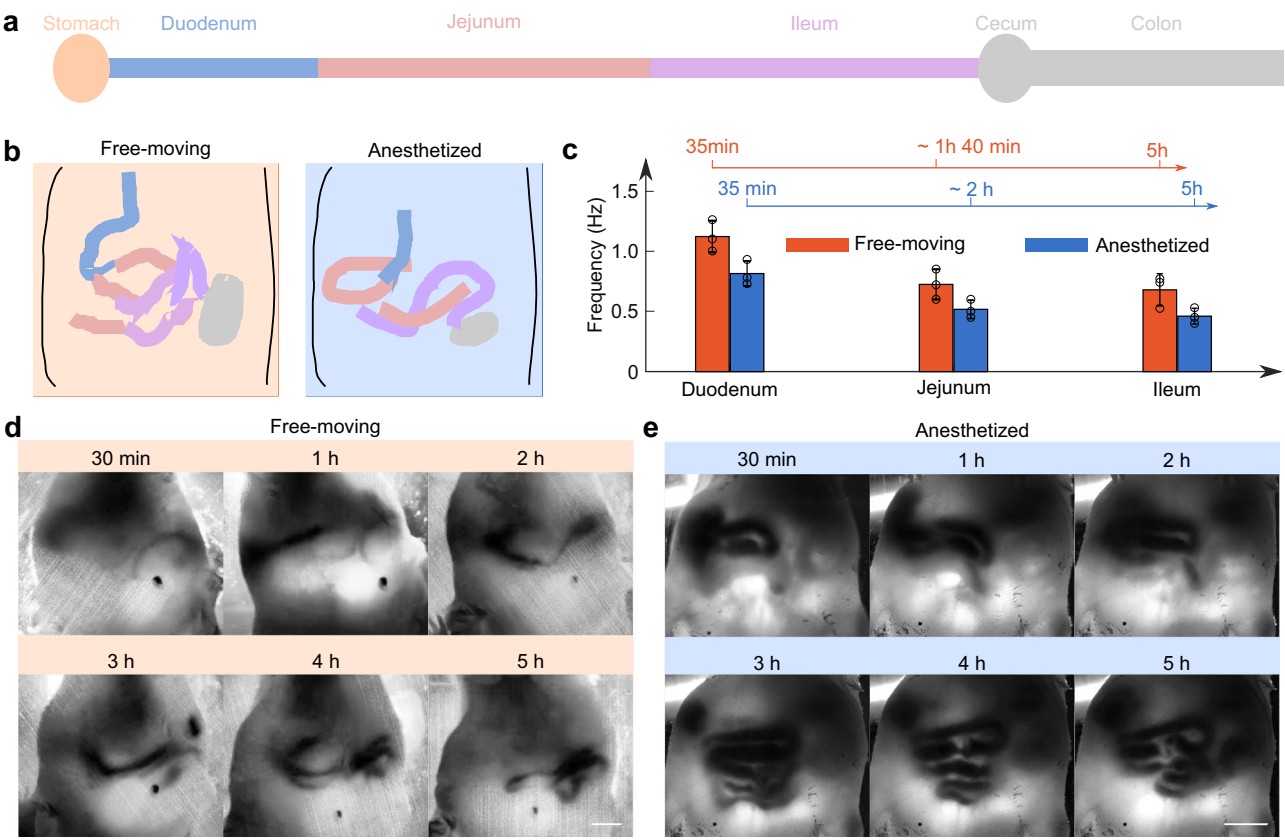

**Fig. 5 Comparison of the frequency of pacemaker-driven myogenic contraction patterns in free-moving and anesthetized mice. a** Schematic drawing shows the structure of intestine. **b** Projected intestine profiles of one free-moving mouse and one anesthetized mouse. **c** A comparison of frequency at different time points for free-moving mice and anesthetized mice. Free-moving mice showed higher motility frequency than the anesthetized mice (mean ± s.d., $n = 3$ mice). **d** The intestine of free-moving mice visualized by TIP at different time points post gavage. Scale bar: 10 mm. **e** The intestine of anesthetized mice visualized by TIP at different time points post gavage. Scale bar: 10 mm.

segmentation motor pattern, that has not been observed in live animals since Cannon showed it with X-rays in the cat, published in 1902, while holding the cat on his lap, with devastating consequences for his health[26]. This study takes away doubt about the importance of slow-wave-driven motor patterns in the small intestine. There is no doubt about the significance of neurally driven peristalsis[24,25], the myogenic, pacemaker-driven activities have received less attention, although they were emphasized by pioneers in intestinal motility research, notably Alvarez in 1924[27]. This may have been in part because low-resolution techniques do not allow fine detail needed to discern patterns such as the Cannon-type segmentation motor pattern[26]. The present study was executed with a non-nutritious meal that will result in a cyclic fasting motor pattern, also termed the migrating motor complex[28,29]. This consists of phase I, which is a quiet phase, and phase II with an "erratic" or fed-state-like activity, followed by phase III that is propulsive in nature. The segmentation motor pattern observed in the present study is likely part of phase II. It can be readily observed in the excised intestine, ex vivo[20]. Strong rhythmic propulsive activity is also observed in the mouse intestine ex vivo, where it has been called migrating motor complex[29] or minute rhythm[30,31]; it is a neurogenic motor activity that can be observed in mutant mice that do not have ICC-MP[29,31]. The minute rhythm occurs both in fasting rats as part of phase II and in fed rats[32]. It will be exciting to apply the technique developed here to mutant mice with selective deletions in classes of enteric neurons[33], deletions in subtypes of ICC assessing primary and stimulus-dependent pacemaker activities[6,23,34], or deletions in glia[35], to determine the role of the various cell types in control of intestinal transit and segmentation.

TIP shows significant advantages over existing modalities. Our system overcomes the invasive nature of an ex vivo study and the requirement of anesthesia in most in vivo modalities, including X-ray imaging, MRI, and photoacoustic imaging[13,36]. The use of two, distinguishable contrast agents is trivial with TIP and can assist in understanding transit, but would be difficult or impossible with X-ray or MRI contrast agents. While fluorescence imaging can potentially be performed on awake animals, it requires a dark environment and in practicality cannot achieve sufficient depth to provide useful measurements. In contrast, our method can be performed under bright conditions, as we rely on the strong transilluminated NIR light instead of the weak fluorescence emission.

TIP will enable intestinal studies that are limited by current pharmacological methods. Pharmacological approaches to gut dysmotilities have had limited success, to a large part because of their serious side effects on the cardiac system. Currently, there is a research emphasis on non-invasive, non-pharmacological techniques[37]. This is accompanied by a strong interest in the initiation and orchestration of gut motor patterns by the autonomic nervous system (ANS)[38]. Although the role of the ANS has been known for a long time, it still does not play a significant role in diagnosis and treatment, largely because animal research over the last 30 years has almost entirely focused on the muscle and enteric nervous system with in vitro experimentation. In addition, certain motor patterns, such as the high-amplitude propagating pressure waves in the human colon[39], do not occur in vitro

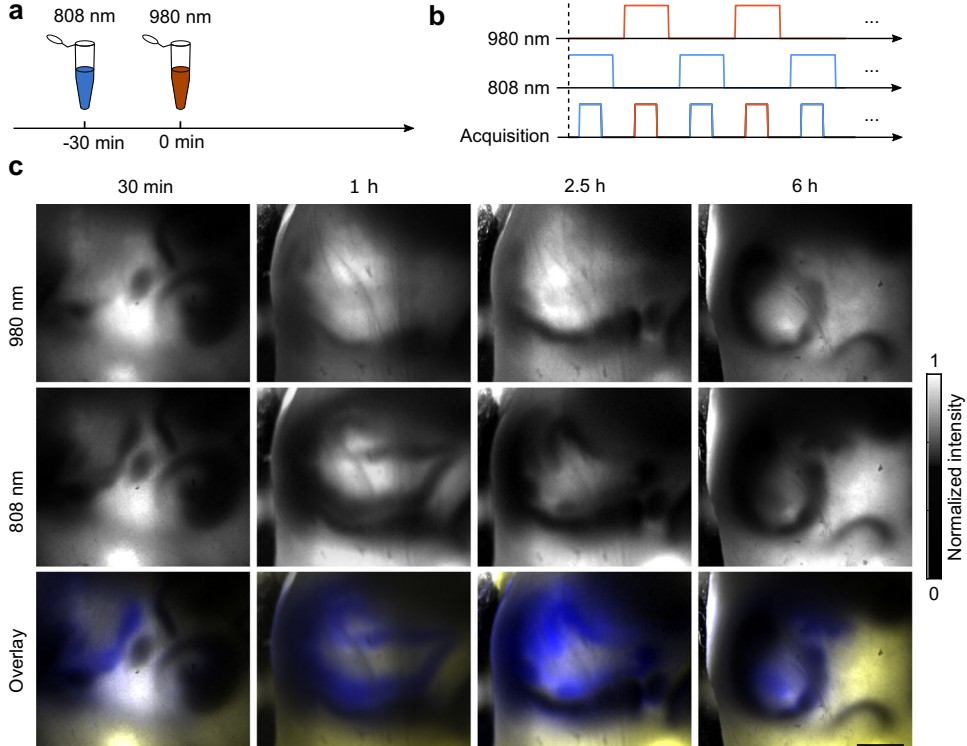

**Fig. 6 Dual-contrast-agent intestinal imaging of mice revealed different sections of the intestine. a** The 808 nm and 980 nm contrast agents were spectrally separate and were gavaged with a 30 min interval. **b** Schematic drawing shows the laser excitation and image acquisition sequences during dual-contrast-agent imaging. **c** The temporal intestinal images acquired at each wavelength (top and middle) and the overlay of the two images (bottom). Scale bar: 10 mm.

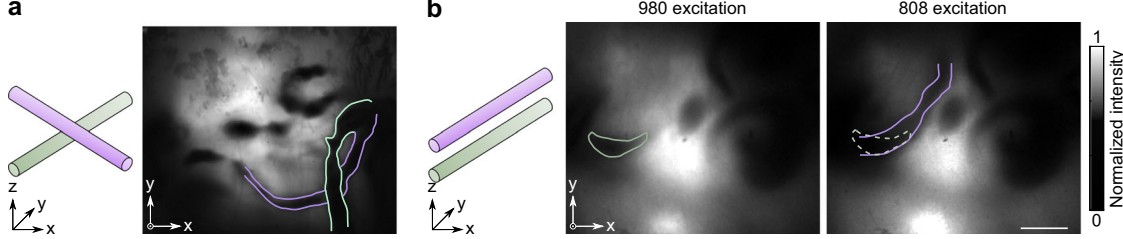

**Fig. 7 TIP resolved overlapped intestinal segments. a** Left: the schematic shows the intersection of two sections of the intestine; right: TIP image clearly showed the intersection of two intestines. **b** Left: the schematic shows the fully overlap of two sections of the intestine; middle and right: dual-contrast TIP imaging differentiated two overlaid intestines. Scale bar: 5 mm.

because they are primarily directed by the autonomic nervous system[40]. Our TIP will be an essential tool to explore studies in vivo. Our TIP is also ideal for translational research into treatment as it has potential to boost research related to non-invasive modulation of the autonomic nervous system at the spinal and vagal level[37].

Unrestrained in vivo imaging of the mouse gastrointestinal system will open up more research possibilities into the role of the central and autonomic nervous systems in gastrointestinal motility physiology and pathophysiology in concert with the myogenically-controlled motor patterns shown in the present study[34]. We hope that our study will spark interest in gut motility research in free-moving animals. In the early 1900s, before the realization of the health risks of exposure to X-rays, several critical studies on live animals came to fruition, but that research soon stopped, never to be fully replaced. We believe that the TIP system has potential to re-vitalize this type of research to better understand the mechanisms of motility control and to discover the pathophysiology of motility disorders.

## Methods

**TIP imaging system**. TIP's excitation and tracking components include a liquid optical fiber bundle (Cone diameter: 5.1 mm, Model: 77636, Newport), a tracking camera (CMUcam5), and an $x$–$y$ motorized translational stage with a 25 cm traveling distance. Through a 3D-printed holder, we mounted the output end of the optical fiber and the tracking camera to the $x$–$y$ translational motorized stage. The optical fiber bundle receives the 808 nm light provided by a laser diode (L808P1000MM, Thorlabs) at the input end, and delivers the light to the mouse's body from the output end. The light beam expands over a distance that encompasses a 3-cm diameter region on the mouse's body (Fig. 1b), leading to a skin surface light intensity of approximately 14 mW/cm, which is far below the safety limit defined by the American National Standards Institute (ANSI) (3.29 W/cm$^2$). The tracking camera was programmed to detect two markers (one yellow and one green) drawn on the mouse's back. Because the FOV of the tracking camera is small, a wide-angle lens (Insignia) was added in front of the camera (Fig. 1a and Supplementary Fig. 1). We also added a low pass filter (700 nm) in front of the wide-angle lens to prevent any reflected laser from entering the tracking camera. The middle region of the two markers is the ideal spot for light illumination (Fig. 1b, d). During the experiment, the coordinates of the markers were transmitted in real-time from the tracking camera to the Raspberry Pi through a USB cable. The tracking camera and the Raspberry Pi work in a closed-loop mode (Fig. 1d): as the animal moves, the coordinates of the two markers change, then the Raspberry Pi calculates the moving distance and sends pulsed signals to the motor

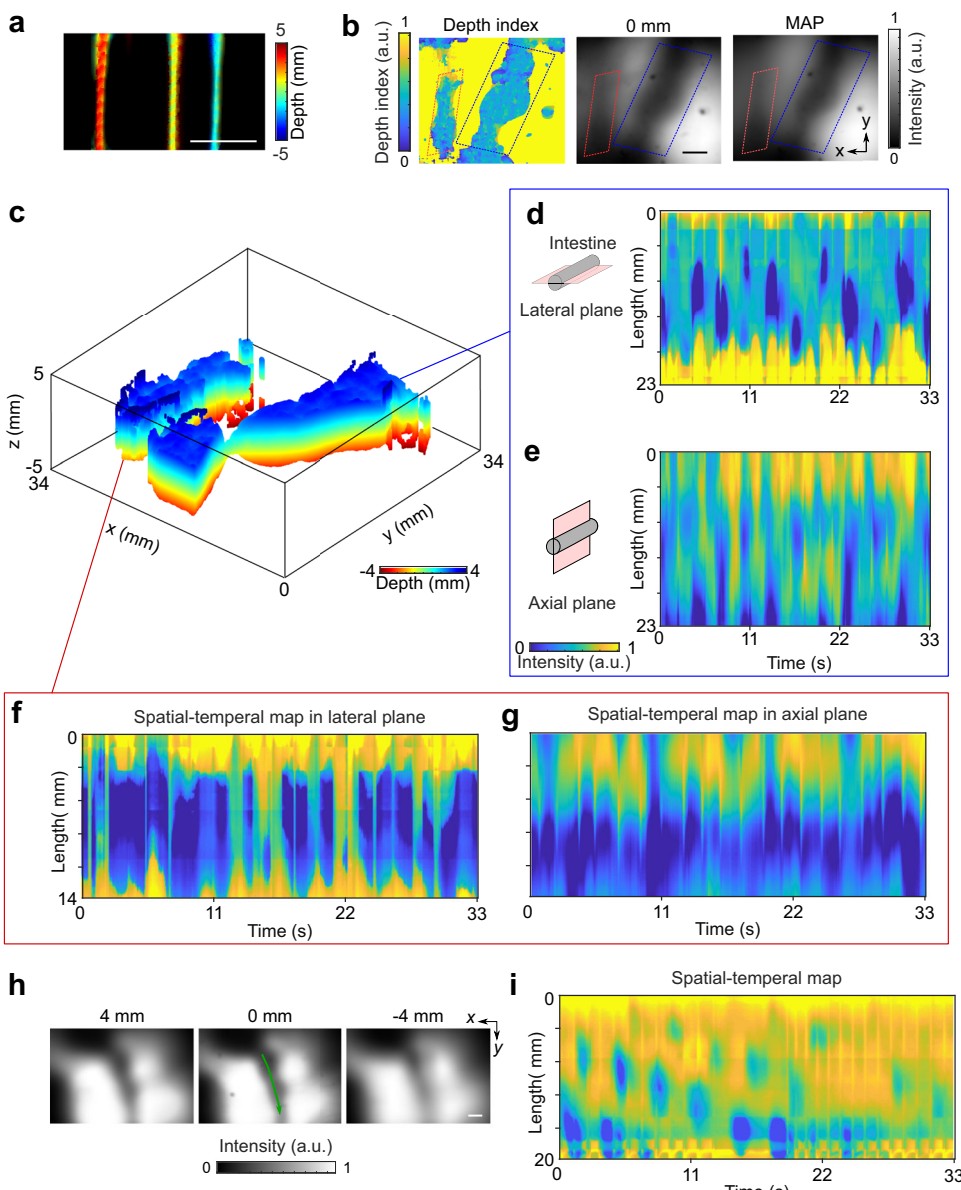

**Fig. 8 3D-TIP of intestine in anesthetized mice. a** 3D-TIP images of three human hairs located at depths of −2, 1, and 5 mm. Scale bar: 1 mm. **b** From left to right: The depth index map of two sections of an intestine; the intestine image at the 0 mm depth; the MAP image of intestines projected along the axial direction. **c** The 3D image of two sections of intestine. **d**, **e** The spatial-temporal maps along the lateral and the axial plane of the intestine labeled within the blue box in (**b**), respectively. **f**, **g** The spatial-temporal maps along the lateral and the axial plane of the intestine labeled within the red box in (**b**), respectively. **h** Intestine images of one mouse (green line shows the selected intestine) over different depths. **i** The spatiotemporal map of the labeled intestine in (**h**). The segmentation motor pattern can be clearly seen. Scale bars in (**b**) and (**h**): 5 mm.

driver, which moves the laser spot back to the central region of the two markers. For imaging, we used a NIR camera (PL-D734MU-NIR-T, Pixel link) placed under the mouse to record the transilluminated NIR light (Fig. 1c and Supplementary Fig. 1). To avoid effects from ambient light, we added a 700 nm long-pass filter (Supplementary Fig. 1) in front of the NIR camera. We also used two perpendicular polarizers, one in front of the fiber output (polarizer 1 in Fig. 1a) and the other in front of the NIR camera (polarizer 2 in Fig. 1a), to block the laser beams that did not interact with the mouse's body.

**Imaging resolution and depth validation**. We quantified the resolution of TIP by imaging a USAF target (USAF 1951 1×, Edmund) under room light without any optical filter set.

We validated the imaging depth through phantom experiments with silicon tubes of two different diameters. The first tube has a 0.5-mm-inner-diameter while the second one has a 4-mm-inner-diameter, which is comparable to the diameter of mouse intestine. We filled the tubes with BNc, which was of a concentration of 100 optical densities (OD), and then imaged the tubes through different thicknesses of agar pieces, which were made with 1% intralipid, 3% agar, 0.2% ink, and 95.8% water to mimic both scattering and absorption (India ink) of biological tissue[41,42].

For the 0.5-mm tube, we imaged it in two modes (Supplementary Fig. 3). One is the fixed $t_1$ mode, where we fixed thickness of the agar phantom placed on top of the tube as 4 mm while increased the thickness of agar phantom below the tube (Supplementary Fig. 3a), and the other is the fixed $t_2$ mode, where we increased the thickness of the agar phantom placed on top of the tube while fixed the thickness of agar below the tube as 4 mm (Supplementary Fig. 3b). To quantitatively analyze the image contrast for tubes, we defined a contrast parameter as shown in Eq. (1).

$$\text{Contrast} = (A_b - A_t)/A_t \qquad (1)$$

Here $A_b$ and $A_t$ are the average amplitudes of the background and tube signals, respectively. Based on Eq. (1), a darker tube and a brighter background result in a larger contrast parameter (Supplementary Fig. 3c). We also quantified the FWHM of the tube (Supplementary Fig. 3d). In the fixed $t_1$ mode, the FWHM of the tube increased quickly and the contrast of the tube decreased rapidly when the thickness of agar under the tube increased, due to the increased scattering below the tube. However, in the fixed $t_2$ mode, the FWHM and the contrast of the tube did not

change significantly. For the experiment with the tube of 4-mm-inner-diameter, we imaged it in the fixed $t_1$ mode, and also quantified the contrast change over different depths (Supplementary Fig. 4). This experiment showed similar results as the 0.5-mm tube imaging.

**Animal preparation**. All animal experiments were performed in accordance with the University at Buffalo's Institutional Animal Care and Use Committee. Six-month-old female ND4 Swiss Webster mice from Envigo were used for all the experiments. The animal facility had a 12-h light/12-h dark cycle, the humidity and temperature were kept at ~50% and 20 °C, respectively. Twenty-four hours before the experiments, the mice fasted with access to only water. On the day of the experiment, we used an electric clipper to shave the hair on each animal's belly and back. Then, we drew a yellow and a green marker on the upper and lower trunk of the mouse back, respectively. Both of these markers are ~1.5 cm in diameter. We then gavaged 200 μL of the contrast agents (concentration: 100 ODs) into each animal and placed it on the imaging platform. We imaged them at 15 Hz frame rate for single-color imaging, and 2 Hz for dual-color imaging.

**Automatic frame registration method**. Because the mouse is moving while the NIR camera stays stationary, we first performed feature-based image registration to the raw video of free-moving imaging, so that the intestinal region of the mouse can be fixed at the same location across all frames. This enabled us to calculate the spatial-temporal map of the intestinal region of interest.

To facilitate image processing, we registered the acquired frames based on the outline of the mouse's body, which occupied ~5% of the camera's FOV (Supplementary Fig. 6). To extract the body outline, we first converted the acquired frame (Supplementary Fig. 6a) into a binary image with an initial threshold of one[43]. The threshold was then reduced by 0.005 iteratively in an iteration loop. In each iteration, the binary image was complemented. As the threshold gradually decreased, the complemented image showed several silhouettes. We focused on the top two largest silhouettes in the frame. The largest silhouette in the frame is the white region in the red box in Supplementary Fig. 6b, which corresponds to a region that is not well illuminated. The mouse area is the second-largest silhouette. Then, the ratio of the mouse region and the FOV of the camera was calculated. If the ratio is lower than 5%, the threshold further decreases and the iteration continues until the ratio reaches 5%. The mouse region was then extracted from the complemented image (Supplementary Fig. 6c).

Next, we computed the convex hull of the mouse silhouette to extract the orientation and centroid features for registration (Supplementary Fig. 6d). The convex hull is used here because it can easily generalize the animal shape. Once the convex hull was obtained, its image moment was calculated. The second-order central moment, which could be used to extract the direction of an image, was utilized to form the covariance matrix[44] (Methods: "Covariance matrix calculation"). Then, the eigenvectors and the corresponding eigenvalues of this matrix were computed, where the largest eigenvalue of the eigenvector denoted the major axis of the convex hull. Following that, the angle ($\theta$) between the major axis and the horizontal axis was calculated, which was the angle we applied to rotate the video frame. The rotation was performed through the transformation matrix $M_R$ (Eq. (2)).

$$M_R = \begin{bmatrix} \cos(90 - \theta) & -\sin(90 - \theta) \\ \sin(90 - \theta) & \cos(90 - \theta) \end{bmatrix} \quad (2)$$

By applying $M_R$ to the original frame, we aligned the intestinal region along a fixed direction (90°) (Supplementary Fig. 6e). After this process, the intestinal region in all the frames was aligned to be upright, and only has displacement. The displacement was then corrected through a translation vector ($t_{ix}, t_{iy}$), which was calculated using Eq. (3).

$$\begin{bmatrix} t_{ix} \\ t_{iy} \end{bmatrix} = \begin{bmatrix} c_{ix} - c_{1x} \\ c_{iy} - c_{1y} \end{bmatrix} \quad (3)$$

Here, ($c_{ix}, c_{iy}$) represents the centroid coordinate ($x, y$) of the mouse silhouette in the current frame ($i_{th}$), while ($c_{1x}, c_{1y}$) is the centroid of the mouse silhouette in the first frame. Using the centroid of the mouse convex hull in the first frame as the anchor point, the mouse silhouette in all the other frames was fixed at the anchor point after a shifting transformation matrix $M_T$ (Eq. (4)) was applied to them (Supplementary Fig. 6f). Thus, the video was successfully registered (Supplementary Fig. 6g). Next, we cropped the region showing clear intestinal motility [region of interest (ROI)] for the spatial-temporal map calculation (Supplementary Fig. 6h).

$$M_T = \begin{bmatrix} 1 & 0 & t_{ix} \\ 0 & 1 & t_{iy} \end{bmatrix} \quad (4)$$

It should be noted that the aforementioned registration method works well for a straight or slightly bent mouse body. When the mouse body was bent severely, the convex hull of the mouse silhouette would be significantly different from that of the straight mouse. The profile of the intestine would also change. Therefore, even if those frames were well registered, it would be difficult to track the same intestine before and after bending for spatiotemporal map quantification. In this case, we

used the similarity based method to compute the intestinal pattern (Fig. 4). We processed data with Matlab (MathWorks).

**Covariance matrix calculation**. The covariance matrix is calculated in the following way. First, we obtained the covariance matrix of the convex hull of the mouse from its second-order central moment, which is calculated using the following equations:

$$\mu'_{xx} = \frac{\mu'_{xx}}{N}, \; \mu'_{yy} = \frac{\mu'_{yy}}{N}, \; \mu'_{xy} = \frac{\mu'_{xy}}{N} \quad (5)$$

where

$$\mu'_{xx} = \frac{\mu'_{xx}}{N}, \; \mu'_{yy} = \frac{\mu'_{yy}}{N}, \; \mu'_{xy} = \frac{\mu'_{xy}}{N} \quad (6)$$

$$\mu'_{yy} = \sum_x \sum_y (y - \bar{y})^2 f(x, y) \quad (7)$$

$$\mu'_{xy} = \sum_x \sum_y (x - \bar{x})(y - \bar{y})f(x, y) \quad (8)$$

Here, $x$ and $y$ are the coordinates of each pixel in the image, ($\bar{x}, \bar{y}$) is the centroid's coordinate of the convex hull, $N$ is its number of pixels inside the convex hull, and $f(x, y)$ is the intensity distribution function of the convex hull. Since the image is binary and the region inside the convex hull is white, $f(x, y)$ equals 1. The covariance matrix ($cov$) of the silhouette is:

$$cov = \begin{bmatrix} \mu'_{xx} & \mu'_{xy} \\ \mu'_{xy} & \mu'_{yy} \end{bmatrix} \quad (9)$$

**Spatial-temporal map**. Using the registered and cropped video (Supplementary Fig. 7a), we calculated the spatial-temporal map of the intestines. First, we enhanced all the frames of the processed video with contrast-limited adaptive histogram equalization (Supplementary Fig. 7b)[45]. Then, we performed the MAP to all the frames in the video. In the MAP image, we manually drew a curve along the intestine of interest (green line in Supplementary Fig. 7c). Because all the frames are registered, the green curve in the MAP image represents the intestine position of the entire video. In each frame, our program automatically calculated the normal vector at each pixel of the curve (red arrows in Supplementary Fig. 7c). All the normal vectors had the same length and each vector fully covered the cross-section of the intestine. Along each normal vector, we extracted the cross-sectional profile of the intestine (Supplementary Fig. 7d). Stacking the profiles of all frames forms a temporal map (Supplementary Fig. 7e). To highlight the intestine regions, we used a global threshold computed using Otsu's method[46] to remove the low amplitude signals in that map (Supplementary Fig. 7f). Each column in Supplementary Fig. 7f corresponds to the cross-sectional profile of intestine at one-time point. This image was further converted into a one-dimensional plot (Supplementary Fig. 7g) by quantifying the summation (or average) of signals in each column. In Supplementary Fig. 7g, valleys represent contractions (smaller intestine cross-sectional diameter: few contrast signals), while peaks indicate relaxations (larger intestine cross-sectional diameter: more contrast signals). We then repeated the same procedure for all cross-sectional vectors in Supplementary Fig. 7d and stack the averaged signals along the intestine direction to form Supplementary Fig. 7h, where each row represents the temporal signal change at one intestine position, while each column represents the spatial signal changes along the intestine at a certain time spot. In this spatial-temporal map, the contractions are shown as spots. We then smoothed the spatial-temporal map with a $2 \times 2$ two-dimensional median filtering.

**Dual-color TIP imaging**. The main BNc micelle contrast agent used for single-color imaging has a peak absorption at 800 nm (BNc) and very low absorbance at 980 nm[3], while the second contrast agent (CyFaP micelle) used with BNc for two-color imaging has high absorption at 980 nm and moderate absorption at 800 nm[47]. Contrast agents were prepared as described in their respective references. A 808-nm laser and a 980-nm laser (L980P200, Thorlabs) were used as respective imaging light sources. For dual-color imaging, we gavaged the 808-nm contrast agent first, and 30 min later we gavaged 980 nm contrast agent. We synchronized the two lasers to illuminate the mouse alternatively, and triggered the camera accordingly to ensure that each frame contained image from only one wavelength. The setup allows us to spectrally resolve the movement of two contrast agents.

**Volumetric intestine imaging**. We imaged intestine volumetrically with a light-field imaging system that was built by replacing the imaging objective of the free-moving system with a lens set containing one macro lens, one MLA (RPCphotonics, MLA-S100-f10) and one 1:1 relay lens. The macro lens images the intestine with a 1/3× magnification ratio. The MLA used for 4D light-field capturing is placed at the focus of the macro lens, and the MLA is then imaged onto the camera sensor with a 1:1 relay lens.

Light-field image reconstruction produced three-dimensional volume from the acquired 2D frame. We first obtained the lenslet parameters from the light-field

images using the guidelines within the Light Field Display software provided by the Stanford Computer Graphics Laboratory[48]. We then applied these parameters to the shift-and-add algorithm to refocus the images at different depths[49,50]. Because we did not deconvolve our 2D images in the free-moving imaging, for comparison we also did not deconvolve our light-field data in 3D imaging.

**Reporting summary**. Further information on research design is available in the Nature Research Reporting Summary linked to this article.

## Data availability

All data generated and analyzed during the current study are contained within the manuscript, and/or are available from the corresponding author on reasonable request.

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

## Acknowledgements

This work is supported in part by the National Institutes of Health (DP5OD017898 and R01 EB029596).

## Author contributions

D.W., J.F.L., and J.X. Conceived the idea. D.W., T.V., Y.Z., A.M., A.R., and S.Z. performed the free-moving experiments. D.W., H.Z., and P.W. performed the light-field experiments. U.C. prepared the contrast agent. D.W., L.W., J.D.H., J.F.L., and J.X. wrote the manuscript.

## Competing interests

The authors declare no competing interests.
