## [Peer Review File · Nature Communications]

REVIEWER COMMENTS

Reviewer #1 (Remarks to the Author):

Capitalizing on near-infrared (NIR) optics and a novel contrast agent, the author's report the first Trans illumination Intestine Projection (TIP) imaging system for free-moving mice.

The study capitalizes using near infrared (NIR) optics and a novel contrast agent, such that TIP is able to identify features located in deep tissue with high imaging contrast. The system overcomes the invasiveness nature of an ex vivo study and the requirement of anesthesia in most in vivo modalities, including X-ray imaging, MRI, and photoacoustic imaging. After a complete evaluation of the system's performance, the authors performed in vivo studies and demonstrated motor patterns using spatial-temporal mapping. The method enables visualization of peristalsis and segmentation motor patterns.

The author's imaged an anesthetized mouse and showed a much slower intestinal motility rate than the awake mouse. It is suggested that TIP system will open up new avenues for functional imaging of the GI tract in animal models. The 2D-TIP was further validated through in vivo imaging of mice at 21 min, 70 min, and 2.5 h post-gavage of contrast agent.

Comments:

This is a very much technology driven study. Whilst the technique clearly has advantages I am not convinced those advantages have been demonstrated. The study records short periods and then spatio-temporal maps are made. The patterns of activity recorded are exactly what has already been well described in the past, even in in vitro recordings from mouse small intestine. I would really like to have seen this technology show something new in terms of data. I do commend the authors for the development of the technique. This aspect is novel. However, I think this manuscript would be better suited to a techniques or methods journal unless major new insights into gut physiology can be presented. There are some major technical hurdles that I believe need to be overcome to ensure this technique could be any better than MRI or X-Ray imaging, which is showing increasing benefits.

Major concerns:

Point 1. The Spatio-temporal maps don't show any recordings of motor activity greater than about 30 seconds. For this technique to be effective and physiologically useful it would be essential to record for the long durations and with point 2 below also taken into account. The fact that the author's don't present any long duration recordings is concerning.

Point 2. This brings onto the next concern is how the technique can keep track of a segment of small bowel that is looped on top of an underlying segment of small bowel. As the author's know, as the animal moves around the cage, the intestines move around the abdominal cavity. How do the author's know this technique can maintain tracking when contrast material is present in different segments of small bowel lying on top of each other (i.e. the dorsal-ventral aspect) ?

Point 3. The light beam expands over a distance that encompasses a 3-cm diameter region on the mouse's body (Fig. 1b), leading to a skin surface light intensity of approximately 14 mW/cm². This is a major concern for longer duration recordings. Recently, in Nature Neuroscience (Owen SF et al. 2019; 22: 1061-1019) it was concluded that: "...commonly used illumination protocols increased the temperature by 0.2-2 °C and suppressed spiking in multiple brain regions." The authors of this Nature Neuroscience paper showed very clearly that minute increases in temperature very powerfully can detrimentally affect the behavior of the mouse. What worries me, is that the authors of this current study use high intensity light illumination to the animal, but there is no evidence that this does not affect behavior. As longer duration recordings are made of gut motility are made, this would of course require longer duration periods of high intensity illumination applied to the animal. This aspect would need to be seriously and thoroughly tested - with appropriate controls. This data was not presented. I would like to suggest for the author's new approach to be really successful, it needs to be able to record for periods a lot longer than 30 second. Transit time in the mouse gut is >20 hours.

Reviewer #2 (Remarks to the Author):

The study by Wang and colleagues aims to develop and validate a novel approach to study gastrointestinal motility in awake and freely moving mice. The approach is to use near-infrared (NIR) imaging, given the depth of penetration, with a gavaged NIR contrast. Then, using a two-camera system – one to track the position of the mouse and the other to record intraabdominal activity – the authors studied the transit of contrast at 21 minutes after gavage. They recorded spatiotemporal activity in an awake and sedated animals and found a decrease in activity in the latter. Then, modifying this system for 3D imaging, they recorded both axial and lateral planes seeing segmentation and peristaltic activities.

The study focuses on gastrointestinal motility, which is an important area of gastrointestinal physiology and pathophysiology. There are currently no approaches that can monitor gastrointestinal motility at high temporal and spatial resolution in an awake mouse. Thus, this would be an important advance. The approach is novel and has solid rationale. However, there are substantial limitations in the conceptualization and study design, data presentation and analysis and interpretations, which substantially limit enthusiasm.

Major:

- The biggest difficulty is that motility is highly variable in amplitude, frequency, and patterns along the length of the gastrointestinal tract. There is a gradient along the length of the small bowel, for example, with duodenum being faster and ileum being slower. Therefore, a 3D “map” of the gastrointestinal tract is required to determine the location of contrast. Comparisons between conditions (e.g. awake and sedated) have to be made in the same gut segments.
- I was hoping that the 3D tracking system may be an approach to reconstruct the intra-abdominal positioning of the intestine to provide location reference (even if the depth of penetration is somewhat shallow). However, the way it is used in the study to map the axial and lateral planes of the intestine does not have a clear relevance. Indeed, the authors did not a clear use for these data.
- The spatial resolution of 1-2 mm is poor. The authors state that “because the mouse intestine has an average diameter of 4 mm, the 1-2 mm spatial resolution is sufficient to image intestine motilities”. This resolution does not allow for spatiotemporal imaging given that contractions are frequently only a small proportion of the intestinal diameter.
- Further, sedation affects locomotion, which is a potential artifact in this approach, further complicating this comparison.
- “Twenty-one minutes post gavaging, most of the gavaged contrast agent has not reached the intestines and only a short section of intestine can be visualized”. Studies have clearly shown that even 30 minutes after liquid gavage, a significant length of the small bowel is filled with the gavaged material (e.g. PMID: 17363407).
- Overall, the presented data on gastrointestinal motility are quite limited, and it was very difficult to make sense of the typical raw data (fig 1 b and c, and fig 2a and c).

Minor

- There are errors in spelling and grammar throughout
- The statement “better extract intestines located far away from the abdomen” does not make sense.

12/31/2020

Responses to the comments on the manuscript “Trans-illumination Intestine Projection imaging of intestinal motility in mice” (Manuscript ID: NCOMMS-20-04089)

We would like to thank you for the insightful comments, criticisms, and suggestions on our manuscript. The reviewers have raised many excellent questions, and we have endeavored to address each issue in detail in this response and in the revised manuscript. In our response, the reviewers’ comments are in **blue font**, and are followed by our detailed responses in black font. Texts which have been newly added or altered in the revised manuscript are in **red font**. These edits match the color of revised text in the completely revised manuscript, which we attach for the reviewers' convenience at the end of this response letter.

Our completely revised manuscript with additional experiments, long-time imaging results, more accurate contrast localization and an extensive physiological focus is a testament to the constructive reviewer comments. We thank the editors and reviewers again for your time and effort on reading and commenting on our manuscript.

Best Regards,

Jun Xia, Ph.D.
Associate Professor
Department of Biomedical Engineering
University at Buffalo

REVIEWER COMMENTS

Reviewer #1 (Remarks to the Author):

Capitalizing on near-infrared (NIR) optics and a novel contrast agent, the author's report the first Trans illumination Intestine Projection (TIP) imaging system for free-moving mice.

The study capitalizes using near infrared (NIR) optics and a novel contrast agent, such that TIP is able to identify features located in deep tissue with high imaging contrast. The system overcomes the invasiveness nature of an ex vivo study and the requirement of anesthesia in most in vivo modalities, including X-ray imaging, MRI, and photoacoustic imaging. After a complete evaluation of the system's performance, the authors performed in vivo studies and demonstrated motor patterns using spatial-temporal mapping. The method enables visualization of peristalsis and segmentation motor patterns.

The author's imaged an anesthetized mouse and showed a much slower intestinal motility rate than the awake mouse. It is suggested that TIP system will open up new avenues for functional imaging of the GI tract in animal models. The 2D-TIP was further validated through in vivo imaging of mice at 21 min, 70 min, and 2.5 h post-gavage of contrast agent.

Authors' response: We thank the reviewer's thoughtful comments and constructive suggestions regarding our work. We have thoroughly revised the manuscript according to the reviewer's suggestions.

Comments:

This is a very much technology driven study. Whilst the technique clearly has advantages I am not convinced those advantages have been demonstrated. The study records short periods and then spatial-

temporal maps are made. The patterns of activity recorded are exactly what has already been well described in the past, even in *in vitro* recordings from mouse small intestine. I would really like to have seen this technology show something new in terms of data. I do commend the authors for the development of the technique. This aspect is novel. However, I think this manuscript would be better suited to a techniques or methods journal unless major new insights into gut physiology can be presented. There are some major technical hurdles that I believe need to be overcome to ensure this technique could be any better than MRI or X-Ray imaging, which is showing increasing benefits.

Authors' response: We thank the reviewer for the rigorous reading!

We agree with the referee that most of the work focuses on the novel imaging methodology, however, many new physiological insights start with new technology. In response to your comments, we have now given the manuscript a strong physiological focus (Physiological advance below and the revised discussion). We also added new data (multi-color imaging results) that has further enhanced TIP's advances. We are not sure if the reviewer was able to watch the movies we have included in the supporting information, but we would appreciate they do so, as the movies reveal the power of this completely novel imaging methodology to unambiguously observe contrast movement in awake and free-moving mice. In comparison, neither X-ray nor MRI has reported intestinal imaging of free-moving mice.

When compared with *in vitro* studies, TIP can reproduce intricate contraction patterns that are revealed *in vitro* because TIP has high resolution and deep imaging depth. In addition, TIP has overwhelming advantages over *in vitro* methods. First, many motor patterns that occur *in vivo* are orchestrated by the autonomic nervous system that is severed by *in vitro* studies, hence translational studies do need *in vivo* studies that are now possible with our technique. Second, TIP has great potential to boost research related to non-invasive modulation of the autonomic nervous system at the spinal and vagal level, which is impossible for *in vitro* studies.

When compared with other intestinal imaging modalities, TIP also has advantages. Instead of simply replicating the well described patterns obtained in previous studies, TIP has one major physiology advance and two technical advances over them.

- **Physiological advance:** For the first time, TIP has demonstrated that the motor patterns controlled by intestinal pacemaker cells are the dominant activity in the mouse intestine under free-moving conditions with contrast fluid. This major finding is a continuation of impactful studies published in *Nature* (Huizinga, *Nature* 1995) and *Nature Communications* (Huizinga, *Nature communications* 2014), setting the next frontier for intestinal study.
- **Technical advances:**
 - (1) For the first time, TIP has extended the intestinal study from the anesthetized state to the free-moving state, which can report more accurate conclusion for diagnostic and therapeutic assessment. TIP achieved this by combining the deep imaging depth of NIR light with the highly roboticized imaging recognition ability of a motion tracking camera. We obtained the benefits of both from this combination: we captured deep tissue intestine image of free-moving mouse that is tracked by the tracking camera for accurate illumination.
 - (2) TIP has demonstrated multi-color imaging that can overcome challenges for other imaging modalities, in particular the challenge in assessment of motility in overlapping sections of the intestine. With matched illumination and contrast agents, in the revised manuscript we have successfully demonstrated TIP's feasibility in multi-color imaging, which localized contrast agents in overlapped intestines. This experiment has further strengthened TIP into an advanced imaging and technology approach.

To further emphasize the novelty, the physiological advance and the great potential of TIP, we also revised the discussion in the revised manuscript.

Corresponding changes in the manuscript

Dual-contrast imaging with TIP

“TIP is also capable of multicolor imaging, by using contrast agents that are spectrally separated in absorbance wavelength. With the matched illumination wavelength, separate contrast agent can be imaged. This would not be straightforward or even possible using other modalities such as X-ray or MR imaging. A major advantage of multicolor TIP is the accurate localization of contrast agents at different sections of intestine, thereby eliminating the problem of overlapping sections of the intestine. As an example, we performed dual-color TIP imaging (setup as described in the Methods). We sequentially gavaged two contrast agents with a 30 min interval (Fig. 6a), and imaged the mouse with two wavelengths, semi-simultaneously (Fig. 6b). Within 2.5 hours, each contrast agent revealed different intestine sections which clearly represented the upper and the downstream sections of the intestine (Fig. 6c). As the contrast agents propagated inside the intestine, they eventually reached the same section of intestine and mixed with each other at the 5 h time point (Fig. 6c). The system can be further modified to enable three-color or four-color imaging, which will offer more possibilities in studying intestine movement. This solves a major problem with MRI or ultrasound imaging that suffers from the inability to visualize or measure motor patterns from overlapping segments of the intestine or colon.”

Fig. 6 Dual-contrast-agent intestinal imaging of mice revealed different sections of the intestine. (a) The 808 nm and 980 nm contrast agents were spectrally separate and were gavaged with a 30 min interval. (b) Schematic drawing shows the laser excitation and image acquisition sequences during dual-contrast-agent imaging. (c) The temporal intestinal images acquired at each wavelength (*top* and *middle*) and the overlay of the two images (*bottom*).

Dual-contrast agent TIP imaging (method)

“The main BNC micelle contrast agent used for single color imaging has a peak absorption at 800 nm (BNC) and very low absorbance at 980 nm³, while the second contrast agent (CyFaP micelle) used with BNC for two-color imaging has high absorption at 980 nm and moderate absorption at 800 nm⁴². Contrast agents were prepared as described in their respective references. A 808 nm laser and a 980 nm laser (L980P200, Thorlabs) were used as respective imaging light sources. For two-color imaging, we gavaged the 808-nm contrast agent first, and 30 min later we gavaged 980 nm contrast agent. We synchronized the two lasers to illuminate the mouse alternatively, and triggered the camera accordingly to ensure that each frame contained image from only one wavelength. The setup allows us to spectrally resolve the movement of two contrast agents.”

Discussion

“In this study ... At different time points post gavage, spatial-temporal maps of the intestine clearly showed the transition of motor patterns over time after filling of the stomach, from peristalsis in the proximal intestine followed by segmentation in the remainder of the intestine to facilitate absorption. We showed that the first motor pattern after gavage is the slow wave driven peristalsis. Our experiments represent the first demonstration of this phenomenon in freely moving mice and it is similar to that observed in restrained mice using X-rays¹⁸. Peristalsis is orchestrated by pacemaker cells, the interstitial cells of Cajal (ICC) associated with the myenteric plexus^{15,16}. Most studies focus on neurally driven motor patterns derived from *in vitro* studies²⁵, but here we show that in free-moving mice peristalsis in the proximal intestine is dominated by ICC pacemaker control. This is followed by segmentation motor patterns, a very characteristic motor pattern controlled by two pacemakers that interact with each other^{22,24}. In the numerous studies on intestinal motor activity, *ex vivo*, this motor pattern is almost never observed, a testament to the essential nature of developing methodology with unrestrained mice. We have also demonstrated that an awake mouse exhibits intestinal motility at a higher frequency compared to an anesthetized mouse, highlighting the control mechanism of the central nervous system. To the best of our knowledge, this study represents the first intestinal imaging in free-moving mice. Future studies will involve the study of motor patterns in the presence of natural content that will induce various neurally driven peristaltic motor patterns^{25,26}.”

Through dual-color TIP imaging, we also demonstrate the feasibility of TIP in differentiating different sections of the intestine. With 3D-TIP, we visualized the volumetric intestine profile and identified an intestine that could be barely seen in 2D imaging and demonstrated the first 3D optical intestine imaging in anesthetized mice. Again, we were able to visualize the segmentation motor pattern, that has not been observed in live animals since Cannon showed it with X-rays in the cat, published in 1902, while holding the cat on his lap, with devastating consequences for his health²⁷. This study takes allays doubt about the importance of slow wave driven motor patterns in the small intestine. Although there is no doubt about the significance of neurally driven peristalsis^{25,26}, the myogenic, pacemaker driven activities have often been diminished, and a crusade to change this started with Alvarez²⁸. Decades of *in vitro* research on neurally driven motor patterns, overshadowed pacemaker driven activities, but here we show a definitive study on free-moving mice, that leaves no doubt about the critical importance of pacemaker driven motor patterns^{29,30}.

TIP shows significant advantages over existing modalities. Our system overcomes the invasive nature of an *ex vivo* study and the requirement of anesthesia in most *in vivo* modalities, including X-ray imaging, MRI, and photoacoustic imaging^{13,31}. The use of two, distinguishable contrast agents is trivial with TIP and can assist in understanding transit, but would be difficult or impossible with X-ray or MRI contrast agents. While fluorescence imaging can potentially be performed on awake animals, it requires a dark environment and in practicality cannot achieve sufficient depth to provide useful measurements. In contrast, our method can be performed under any light conditions, as we rely on the strong trans-illuminated NIR light instead of the weak fluorescence emission.

TIP will enable intestinal studies that are limited by current pharmacological methods. Pharmacological approaches to gut dysmotilities have had limited success, to a large part because of their serious side effects on the cardiac system. Currently, there is a research emphasis on non-invasive, non-pharmacological techniques³². This is accompanied by a strong interest in the initiation and orchestration of gut motor patterns by the autonomic nervous system (ANS)³³. Although the role of the ANS has been known for a long time, it still does not play a significant role in diagnosis and treatment, to a large part because animal research over the last 30 years has almost entirely focused on the muscle and enteric nervous system with *in vitro* experimentation. In addition, certain motor patterns, such as the high-amplitude propagating pressure waves in the human colon³⁴, do not occur *in vitro* because they are primarily directed by the autonomic nervous system³⁵. Our TIP will be an essential tool to explore studies *in vivo*. Our TIP is also ideal for translational research into treatment as it has potential to boost research related to non-invasive modulation of the autonomic nervous system at the spinal and vagal level³².

Unrestrained *in vivo* imaging of the mouse gastrointestinal system will open up new research possibilities into the role of the central and autonomic nervous systems in gastrointestinal motility physiology and pathophysiology in concert with the myogenically-controlled motor patterns shown in the present study²⁹. We hope that our

study will spark interest in gut motility research in free-moving animals. In the early 1900s, before the realization of the health risks of exposure to X-rays, several critical studies on live animals came to fruition, but that research soon stopped, never to be fully replaced. We believe that the TIP system has potential to re-vitalize this type of research to better understand the mechanisms of motility control and to discover the pathophysiology of motility disorders.”

Major concerns:

Point 1. The Spatio-temporal maps don't show any recordings of motor activity greater than about 30 seconds. For this technique to be effective and physiologically useful it would be essential to record for the long durations and with point 2 below also taken into account. The fact that the author's don't present any long duration recordings is concerning.

Authors' response: We thank the reviewer for the comment. TIP can perform long-duration imaging. We have now added longer recording results in the revised manuscript - we provided a spatial-temporal map with a duration of up to 12 min (Supplementary Fig. 13). We also extended our frequency comparison to 5 h, which is more than enough to study effects of meal intake and similar studies. Beyond this time point, most contrast agents would reach the cecum and colon¹⁹.

Corresponding changes in the manuscript

Assessment of TIP for long-duration imaging

“Long-duration imaging is important for continuous monitoring of intestinal motility and changing motor patterns, however, there are challenges. A potential concern is ... Another challenge is that the distribution of the intestine inside the abdominal cavity will change while the mouse is moving freely. These changes might affect the extraction of the intestine for spatial-temporal map calculation. To assess whether TIP can overcome these challenges, we performed two tests. In our first test, we ... Our second test was to extract the intestine profiles from TIP images involving different intestine distributions (Fig. 4a). Although the entire intestine moves in response to different animal behaviors, TIP could clearly capture the intestine due to the high absorbance of the contrast agent. To digitally extract the intestine, we first calculated the similarity of all frames to separate different behaviors (Fig. 4b, Supplementary Fig. 13a). We then extracted the intestine under each behavior and computed the spatial-temporal map. We combined all spatial-temporal maps to form a long-time map (Fig. 4c and Supplementary Fig. 13b), which proves that TIP is capable of long-duration imaging. The calculation of

long-duration motor pattern for anesthetized mice is easier and does not require tracking of the moving intestine (Fig. 4d and Supplementary Fig. 13c).”

Fig. 4 TIP demonstrated long-duration imaging of mouse intestine. (a) TIP captured the profile of the mouse intestine when the mouse performed different behaviors in a free-moving state. **(b)** The similarity of frames acquired from the free-moving mouse. The similarity changed when the mouse changed its behavior. **(c)** Similarity-based data processing generated the long-duration spatial-temporal map for the free-moving mouse. To show the details of the sub pattern, only three sections of the whole pattern are shown here. A combined full pattern can be seen in Supplementary Fig 13a. **(d)** The spatial-temporal map of the intestine of an anesthetized mouse, acquired over 6 min. White circles indicate intestinal contractions.

Supplementary Figure 13: (a) A combined pattern for mouse with different behaviors over 6 minutes. Longer duration spatial-temporal map for (b) free-moving mouse and (c) anesthetized mouse.

“Once we obtained the motor patterns, we compared the intestinal motility between free-moving mice and anesthetized mice. For better comparison, we also projected intestines imaged at different time points together to form a map of all the visualized intestines over a time window of 5 h. We identified intestines using the stomach and cecum as landmarks – the duodenum is closer to the stomach, the ileum is connected to the cecum, and the intestine in the middle is jejunum (Fig. 5a). For either anesthetized or free-moving mice, we compared the dominant contraction frequency for the same intestine section (Fig. 5b). The duodenum appeared ~30 min post gavage for both the free-moving mice and the anesthetized mice, and the jejunum showed up 1.5 h post gavage. We continued imaging the mice up to 5 h post-gavage. Over the imaged time window, both the free-moving mice and the anesthetized mice showed a gradually decreased motility frequency over time, as the contrast moved from duodenum to ileum. This exhibited the intrinsic frequency gradient of the pacemaker activity, which is essential for anal propagation. Compared to the free-moving mouse, the anesthetized mouse exhibited a lower motility frequency for the same intestinal section (Fig. 5c)²². For both the free-moving mice and the anesthetized mice, TIP visualized the intestine filled by the contrast agent at different time points (Fig. 5d-e), providing a panoramic view of the intestine.”

Fig. 5 Comparison of the frequency of pacemaker driven myogenic contraction patterns in free-moving and anesthetized mice. (a) Schematic drawing shows the structure of intestine. (b) Projected intestine profiles of one free-moving mouse and one anesthetized mouse. (c) A comparison of frequency at different time points for free-moving mice and anesthetized mice. Free-moving mice showed higher motility frequency than the anesthetized mice, (mean \pm s.d., $n = 3$ mice). (d) The intestine of free-moving mice visualized by TIP at different time points post-gavage. Scale bar: 10 mm. (e) The intestine of anesthetized mice visualized by TIP at different time points post-gavage. Scale bar: 10 mm.

Point 2. This brings onto the next concern is how the technique can keep track of a segment of small bowel that is looped on top of an underlying segment of small bowel. As the author's know, as the animal moves around the cage, the intestines move around the abdominal cavity. How do the author's know this technique can maintain tracking when contrast material is present in different segments of small bowel lying on top of each other (i.e. the dorsal-ventral aspect)?

Authors' response: We thank the reviewer for the critical comment. We have multiple strategies to combat intestine overlap and movement during imaging.

First, to ensure that the free-moving mouse is within the depth-of-field of the imaging camera, we designed a transparent box that is only slightly higher than the height of mouse body to restrict the mouse within the box. The height of the box allows the mouse to move freely, while lightly stretching the mouse body over the lateral direction to reduce the overlap of intestine.

Second, we classified the overlap of intestine into two cases. One case is “low overlap” where the two sections of intestine form a cross with one on top of the other. In this case, we show that the two intestines can be visualized with our TIP. The other case is “high overlap” where one intestine is right above the other and is fully covered during imaging. For this case, we added a new experiment that used two contrast agents that were spectrally separable. With matched laser excitation, the highly overlapped two sections of intestines could be separated.

Thirdly, we agree with the reviewer that the intestine moves inside the abdominal cavity in free-moving imaging, but due to the high absorbance of the contrast agent and the low scattering in the near-infrared window, TIP can optically visualize the intestine when mice were performing different behaviors, including turning left, moving forward, turning right and bending its body. All of these behaviors are common actions that a mouse would perform when moving around the imaging cage. Then, with our similarity-based segmentation method, we quantified the sub spatial-temporal map for the segmented videos that correspond to different mouse behaviors. We montaged different sub spatial-temporal maps through correlation. For the results, please refer to our response for point 1 of Reviewer 1.

Corresponding changes in the manuscript

TIP resolved overlapped intestines

“The small intestine of the mouse always has overlapping segments that prevent proper *in vivo* studies of propulsive movements using current techniques. We

demonstrate here that TIP resolved overlapped intestines in single-color and dual-color imaging. We classified the overlap of intestine into two cases. One case showed “low overlap” where the two sections of intestine formed a cross with one on top of the other. In this case, we show that the two intestines can be visualized with our TIP (Fig. 7 a). The other case showed “high overlap” where one intestine was right above the other and was fully covered during imaging. In this case, our dual-color TIP imaging easily differentiated the highly overlapped two sections of intestines (Fig. 7b). Overlapping intestinal segments are one major reason why *in vivo* studies have not been widely explored, but this problem is now solved by our TIP.”

Fig. 7 TIP resolved overlapped intestinal segments. (a) *Left*: the schematic shows the intersection of two sections of the intestine; *right*: TIP image clearly showed the intersection of two intestines. (b) *Left*: the schematic shows the fully overlap of two sections of the intestine; *middle and right*: dual-contrast TIP imaging differentiated two overlaid intestines.

Point 3. The light beam expands over a distance that encompasses a 3-cm diameter region on the mouse’s body (Fig. 1b), leading to a skin surface light intensity of approximately 14 mW/cm². This is a major concern for longer duration recordings. Recently, in Nature Neuroscience (Owen SF et al. 2019; 22: 1061-1019) it was concluded that: “..commonly used illumination protocols increased the temperature by 0.2-2 °C and suppressed spiking in multiple brain regions.” The authors of this Nature Neuroscience paper showed very clearly that minute increases in temperature very powerfully can detrimentally affect the behavior of the mouse. What worries me, is that the authors of this current study use high intensity light illumination to the animal, but there is no evidence that this does not affect behavior. As longer duration recordings are made of gut motility are made, this would of course require longer duration periods of high intensity illumination applied to the animal. This aspect would need to be seriously and thoroughly tested - with appropriate controls. This data was not presented. I would like to suggest for the

author's new approach to be really successful, it needs to be able to record for periods a lot longer than 30 second. Transit time in the mouse gut is >20 hours.

Authors' response: We agree with the reviewer that high illumination light intensity will affect animal's behavior, as has been investigated in the reference ^{20,21}. However, laser safety is determined by the light intensity instead of the total power. In the referenced study [20], while the total power is only 3 mW, the light intensity is 9554 mW/cm², which is 682 times higher than the 14 mW/cm² intensity used in our study. In fact, our light intensity is much lower than the maximum permissible skin exposure limit set by the American National Standards Institute (~300 mW/cm²).

To test that the light intensity used in our study will not cause a temperature rise in animals, we illuminated a piece of chicken breast tissue, a widely used phantom to mimic the optical property of a live animal, with a light intensity of 14 mW/cm² for 3 hours at room temperature and measured the temperature change with a thermal camera. As a control, we also imaged another piece of chicken breast tissue without laser illumination. As expected, we did not detect a temperature rise over time in the illuminated tissue and we also did not discover a temperature difference between the illuminated and the control tissues.

We now performed long-duration (5 h) recordings and have extended our analysis to 5 h, a duration long enough for any intervention study. Beyond this time point, most contrast agent would have reached cecum and colon ¹⁹. Please refer to our response for Point 1 of Reviewer 1.

Corresponding changes in the manuscript

“Long-duration imaging is important for continuous monitoring of intestinal motility and changing motor patterns, however, there are challenges. A potential concern is that long-time illumination of a mouse with the light intensity required by TIP might cause a temperature rise in the mouse body that would affect mouse behavior, which could prevent long time recording. Another challenge is ... To assess whether TIP can overcome these challenges, we performed two tests. In our first test, we illuminated a piece of chicken breast tissue for 2 h at a laser intensity of 14 mW/cm² (the same intensity used in the imaging experiment) and continuously monitored the surface temperature using a thermal camera (FLIR one). As a control, we also imaged another

piece of chicken breast tissue without laser illumination. The result indicated that there was no temperature rise in the exposed tissue and no difference between the exposed and control tissues (Supplementary Fig. 12), proving that long time illumination is not a concern for TIP, at least from a photothermal hyperthermia perspective.”

Supplementary Figure 12: Thermal images of the control and illuminated chicken breast tissue over 2 hours. Compared with the control, the laser power used in the study did not cause a temperature rise in the illuminated chicken breast tissue.

Reviewer #2 (Remarks to the Author):

The study by Wang and colleagues aims to develop and validate a novel approach to study gastrointestinal motility in awake and freely moving mice. The approach is to use near-infrared (NIR) imaging, given the depth of penetration, with a gavaged NIR contrast. Then, using a two-camera system – one to track the position of the mouse and the other to record intraabdominal activity – the authors studied the transit of contrast at 21 minutes after gavage. They recorded spatiotemporal activity in an awake and sedated animals and found a decrease in activity in the latter. Then, modifying this system for 3D imaging, they recorded both axial and lateral planes seeing segmentation and peristaltic activities.

The study focuses on gastrointestinal motility, which is an important area of gastrointestinal physiology and pathophysiology. There are currently no approaches that can monitor gastrointestinal motility at high temporal and spatial resolution in an awake mouse. Thus, this would be an important advance. The approach is novel and has solid rationale. However, there are substantial limitations in the conceptualization and study design, data presentation and analysis and interpretations, which substantially limit enthusiasm.

Authors' response: We thank the reviewer's thoughtful comments and constructive suggestions regarding our work! We have revised the manuscript according to the reviewer's suggestion.

Major:

Point 1: The biggest difficulty is that motility is highly variable in amplitude, frequency, and patterns

along the length of the gastrointestinal tract. There is a gradient along the length of the small bowel, for example, with duodenum being faster and ileum being slower. Therefore, a 3D “map” of the gastrointestinal tract is required to determine the location of contrast. Comparisons between conditions (e.g. awake and sedated) have to be made in the same gut segments.

Authors’ response: We thank the reviewer for the comment. By projecting the intestines imaged at different time points, we have now reconstructed an extended length of intestine and compared the dynamics for the same gut segment between anesthesia mice and free-moving mice. We also compared the motility frequency between free-moving mice and anesthetized mice for the same intestinal section.

Corresponding changes in the manuscript

“Once we obtained the motor patterns, we compared the intestinal motility between free-moving mice and anesthetized mice. For better comparison, we also projected intestines imaged at different time points together to form a map of all the visualized intestines over a time window of 5 h. We identified intestines using the stomach and cecum as landmarks – the duodenum is closer to the stomach, the ileum is connected to the cecum, and the intestine in the middle is jejunum (Fig. 5a). For either anesthetized or free-moving mice, we compared the dominant contraction frequency for the same intestine section (Fig. 5b). The duodenum appeared ~30 min post gavage for both the free-moving mice and the anesthetized mice, and the jejunum showed up 1.5 h post gavage. We continued imaging the mice up to 5 h post-gavage. Over the imaged time window, both the free-moving mice and the anesthetized mice showed a gradually decreased motility frequency over time, as the contrast moved from duodenum to ileum. This exhibited the intrinsic frequency gradient of the pacemaker activity, which is essential for anal propagation. Compared to the free-moving mouse, the anesthetized mouse exhibited a lower motility frequency for the same intestinal section (Fig. 5c)²². For both the free-moving mice and the anesthetized mice, TIP visualized the intestine filled by the contrast agent at different time points (Fig. 5d-e), providing a panoramic view of the intestine.”

Fig. 5 Comparison of the frequency of pacemaker driven myogenic contraction patterns in free-moving and anesthetized mice. (a) Schematic drawing shows the structure of intestine. (b) Projected intestine profiles of one free-moving mouse and one anesthetized mouse. (c) A comparison of frequency at different time points for free-moving mice and anesthetized mice. Free-moving mice showed higher motility frequency than the anesthetized mice, (mean \pm s.d., $n = 3$ mice). (d) The intestine of free-moving mice visualized by TIP at different time points post-gavage. Scale bar: 10 mm. (e) The intestine of anesthetized mice visualized by TIP at different time points post-gavage. Scale bar: 10 mm.

Point 2: I was hoping that the 3D tracking system may be an approach to reconstruct the intra-abdominal positioning of the intestine to provide location reference (even if the depth of penetration is somewhat shallow). However, the way it is used in the study to map the axial and lateral planes of the intestine does not have a clear relevance. Indeed, the authors did not a clear use for these data.

Authors' response: We thank the reviewer for this insight. We have now performed another light-field experiment with longer imaging duration, which allowed us to generate a map of intestine distribution with depth information. We have now added the imaging results in the revised manuscript.

A major use of TIP is to read out the motility of intestine at high spatial and temporal resolution, which is crucial for the determination of the type of motor pattern that is needed for further analysis and diagnosis. Driven by this, we computed the motor pattern using the 3D data. We have now highlighted this information in the revised discussion.

Corresponding changes in the manuscript

“To image movements along the entire length of the intestine, we imaged the mice over 5 h and acquired data at different time points post gavage of contrast agent (Supplementary Fig. 17). Similar to 2D imaging, 3D-TIP visualized sections of intestine over time. To display the depth information, we overlaid the depth index of intestine on top of the intestine image at the principal focal plane (Supplementary Fig. 16). Similar to 2D imaging, we observed more and more intestine sections as the contrast agent moved inside the intestine.”

Supplementary Figure 16: Long-duration 3D-TIP imaging of intestine allowed the generation of intestinal maps with depth information. The gray images show the intestine profile and the color images show the depth index of the intestine. Scale bar: 7 mm.

Discussion

“In this study ... At different time points post gavage, spatial-temporal maps of the intestine clearly showed the transition of motor patterns over time after filling of the stomach, from peristalsis in the proximal intestine followed by segmentation in the remainder of the intestine to facilitate absorption. We showed that the first motor pattern after gavage is the slow wave driven peristalsis. Our experiments represent the first

demonstration of this phenomenon in freely moving mice and it is similar to that observed in restrained mice using X-rays¹⁸. Peristalsis is orchestrated by pacemaker cells, the interstitial cells of Cajal (ICC) associated with the myenteric plexus^{15,16}. Most studies focus on neurally driven motor patterns derived from *in vitro* studies²⁵, but here we show that in free-moving mice peristalsis in the proximal intestine is dominated by ICC pacemaker control. This is followed by segmentation motor patterns, a very characteristic motor pattern controlled by two pacemakers that interact with each other^{22,24}. In the numerous studies on intestinal motor activity, *ex vivo*, this motor pattern is almost never observed, a testament to the essential nature of developing methodology with unrestrained mice. We have also demonstrated that an awake mouse exhibits intestinal motility at a higher frequency compared to an anesthetized mouse, highlighting the control mechanism of the central nervous system. To the best of our knowledge, this study represents the first intestinal imaging in free-moving mice. Future studies will involve the study of motor patterns in the presence of natural content that will induce various neurally driven peristaltic motor patterns^{25,26}.

Through dual-color TIP imaging, we also demonstrate the feasibility of TIP in differentiating different sections of the intestine. With 3D-TIP, we visualized the volumetric intestine profile and identified an intestine that could be barely seen in 2D imaging and demonstrated the first 3D optical intestine imaging in anesthetized mice. Again, we were able to visualize the segmentation motor pattern, that has not been observed in live animals since Cannon showed it with X-rays in the cat, published in 1902, while holding the cat on his lap, with devastating consequences for his health²⁷. This study takes allays doubt about the importance of slow wave driven motor patterns in the small intestine. Although there is no doubt about the significance of neurally driven peristalsis^{25,26}, the myogenic, pacemaker driven activities have often been diminished, and a crusade to change this started with Alvarez²⁸. Decades of *in vitro* research on neurally driven motor patterns, overshadowed pacemaker driven activities, but here we show a definitive study on free-moving mice, that leaves no doubt about the critical importance of pacemaker driven motor patterns^{29,30}.

TIP shows significant advantages over existing modalities. Our system overcomes the invasive nature of an *ex vivo* study and the requirement of anesthesia in most *in vivo* modalities, including X-ray imaging, MRI, and photoacoustic imaging^{13,31}. The use of two, distinguishable contrast agents is trivial with TIP and can assist in understanding transit, but would be difficult or impossible with X-ray or MRI contrast agents. While fluorescence imaging can potentially be performed on awake animals, it requires a dark environment and in practicality cannot achieve sufficient depth to provide useful measurements. In contrast, our method can be performed under any light conditions, as we rely on the strong trans-illuminated NIR light instead of the weak fluorescence emission.

TIP will enable intestinal studies that are limited by current pharmacological methods. Pharmacological approaches to gut dysmotilities have had limited success, to a large part because of their serious side effects on the cardiac system. Currently, there is a research emphasis on non-invasive, non-pharmacological techniques³². This is

accompanied by a strong interest in the initiation and orchestration of gut motor patterns by the autonomic nervous system (ANS) ³³. Although the role of the ANS has been known for a long time, it still does not play a significant role in diagnosis and treatment, to a large part because animal research over the last 30 years has almost entirely focused on the muscle and enteric nervous system with *in vitro* experimentation. In addition, certain motor patterns, such as the high-amplitude propagating pressure waves in the human colon ³⁴, do not occur *in vitro* because they are primarily directed by the autonomic nervous system ³⁵. Our TIP will be an essential tool to explore studies *in vivo*. Our TIP is also ideal for translational research into treatment as it has potential to boost research related to non-invasive modulation of the autonomic nervous system at the spinal and vagal level ³².

Unrestrained *in vivo* imaging of the mouse gastrointestinal system will open up new research possibilities into the role of the central and autonomic nervous systems in gastrointestinal motility physiology and pathophysiology in concert with the myogenically-controlled motor patterns shown in the present study ²⁹. We hope that our study will spark interest in gut motility research in free-moving animals. In the early 1900s, before the realization of the health risks of exposure to X-rays, several critical studies on live animals came to fruition, but that research soon stopped, never to be fully replaced. We believe that the TIP system has potential to re-vitalize this type of research to better understand the mechanisms of motility control and to discover the pathophysiology of motility disorders.”

Point 3: The spatial resolution of 1-2 mm is poor. The authors state that “because the mouse intestine has an average diameter of 4 mm, the 1-2 mm spatial resolution is sufficient to image intestine motilities”. This resolution does not allow for spatiotemporal imaging given that contractions are frequently only a small proportion of the intestinal diameter.

Authors’ response: Based on our observation, contraction will induce large changes (75%) in intestine diameter (Supplementary Fig. 5). For intestine with an averaged diameter of 4 mm, the change corresponds to 3 mm, which is much larger than our spatial resolution. We have now explained this point in the revised manuscript.

In addition, 1-2 mm is spatial resolution quantified in tissue-mimicking media. To quantify the resolution in a scattering-free medium, we imaged a USAF target and characterized the resolution systematically. We verified that our system could resolve element 3 in group 2 of the USAF resolution target, yielding a resolution of 99.2 μm .

Corresponding changes in the manuscript

“To verify the imaging resolution of TIP in scattering-free medium, we imaged a United States Air Force (USAF) resolution target in air and quantified a resolution of $99.2\ \mu\text{m}$ (Supplementary Fig. 2). To verify the imaging depth of TIP, we imaged BNC-filled tubes embedded in agar gels, which mimic both absorption and scattering of biological tissue (Supplementary Figs. 3&4 and Methods). The results indicate that TIP can visualize the 0.5 mm-inner-diameter tube at up to 8 mm depth. This imaging depth covers the majority of intestines underneath the abdomen wall¹³. The mouse intestine has an average diameter of 4 mm¹⁴ in the resting condition, but the diameter dramatically decreases during contraction. We experimentally validated this by quantifying the diameter of a contracted intestine which is 25% of the 4 mm averaged intestine diameter (Supplementary Fig. 5). This observation suggested a 75% change in intestine diameter (3 mm), which is large enough for TIP to capture the changes.”

Supplementary Figure 2 Resolution quantification using the USAF target demonstrated that 2D-TIP offered a lateral resolution of $99.2\ \mu\text{m}$. (a) An image of the USAF target captured with 2D-TIP. (b) The intensity profile along the dashed line in (a) shows that 2D-TIP clearly resolved elements in group 2.

Supplementary Figure 5 TIP identified the contracted intestine. (a) Image of the intestine. The dashed line labeled the cross-section of the contracted intestine. (b) The distribution of the intensity profile of the labeled intestine cross-section in (a). The curve is flipped

upside down for display purposes. The contracted intestine only covered around 25% of the intestine diameter.

Imaging Resolution and Depth Validation (Method)

“We quantified the resolution of TIP by imaging a USAF target (USAF 1951 1×, Edmund) under room light without any optical filter set.”

Point 4: Further, sedation affects locomotion, which is a potential artifact in this approach, further complicating this comparison.

Authors' response: We thank the review for the comment! We want to emphasize that the major application of TIP is imaging of free-moving mice. We also want to highlight that the main focus of this study is introducing the new imaging technique, rather than comparing the intestinal dynamics between anesthetized mice and the free-moving mice. We have now revised the abstract and the discussion in the manuscript to further highlight the innovation of our technique. In addition, we have placed new emphasis on the new physiological insights our study has obtained. Please see the changes in abstract below, and please see the revised discussion in our response for point 2 of Reviewer 2.

We also agree with the reviewer that sedation will affect intestinal motility, but our TIP can still enable us to compare the intestinal motility between anesthetized mice and free-moving mice. For the results, please refer to our response for point 1 of Reviewer 2.

Corresponding changes in the manuscript

Abstract

“The method enables visualization of peristalsis and segmentation motor patterns of unrestrained and unanesthetized mice. We show here that motor patterns controlled by intestinal pacemaker cells dominate activity evoked by distention due to the contrast fluid. We also show the effects of anesthesia on motor patterns, highlighting the role of the extrinsic nervous system in controlling motor patterns. Combining with light-field technologies, we further demonstrated 3D optical imaging of intestine *in vivo* (3D-TIP), providing evidence for the hypothesis that a slow moving motor pattern is underlying the generation of segmentation. Importantly, the added depth information allows us to extract intestines located away from the abdominal wall, and to identify and quantify intestinal

motor patterns along different directions. The TIP system should open up new avenues for functional imaging of the GI tract **in conscious animals in natural physiological states.**”

Point 5: “Twenty-one minutes post gavaging, most of the gavaged contrast agent has not reached the intestines and only a short section of intestine can be visualized”. Studies have clearly shown that even 30 minutes after liquid gavage, a significant length of the small bowel is filled with the gavaged material (e.g. PMID: 17363407).

Authors’ response: We thank the reviewer for the critical comment. We agree with the reviewer that the gastric emptying can be high at 30 min post the gavage, as shown in reference ²³. Our TIP also observed that a decent length of intestine was filled with contrast agent 30 min post gavage (Supplementary Figure 8). Based on these findings, we have now rephrased the text in the manuscript into “**Twenty-one minutes post gavaging, we observed that the intestine started to be filled with contrast agent**”, which we believe is more accurate.

Corresponding changes in the manuscript

“**Twenty-one min post-gavage, 2D-TIP showed that the intestine started to be filled with contrast. (Fig. 2a: top). At this stage, the dominant intestinal motility is peristalsis driven slow wave, which originates from the pacemaker cells and the intersitital cells of Cajal ^{15,16}. The peristaltic activity is a wavelike movement that pushes the contrast agent forward and is shown as propagating bands in the spatial-temporal map (Fig. 2b: top) ^{17,18}.**

Thirty min post gavage, 2D-TIP showed that more sections of intestine became filled with contrast (Supplementary Fig 8), consistent with previous study ¹⁹.”

Supplementary Figure 8 TIP revealed intestine structure 30 min post the gavage.

Point 6: Overall, the presented data on gastrointestinal motility are quite limited, and it was very difficult to make sense of the typical raw data (fig 1 b and c, and fig 2a and c).

Authors response: We thank the reviewer for this comment. To better convey information of the raw data, we have now added a new figure (Fig. 3) in the manuscript. Panel (a) of the new figure contains raw data that shows the contrast agent movement in the intestine over time when driven by the peristaltic motor pattern. Panel (b) of the new figure shows the movement of contrast agent inside intestine under the driven of segmentation motor pattern. We hope the figure provides helpful information for the reviewer and readers to understand the raw data.

We are not sure if the reviewer was able to watch the movies we have included in the supporting information, but we would appreciate they do so, as in the movies, it is quite unambiguous and straightforward to observe the contrast movement.

Corresponding changes in the manuscript

“To better reveal the movement of the intestine, we show its activity at different times, which clearly represents the movement of contrast agent in the intestine. Within a time window of 120 s, we observed pacemaker-driven peristalsis, which causes propulsion to move content in the anal direction²². Due to peristalsis, the contrast agent moved rapidly over a distance of more than 20 mm, filling most sections of the intestine shown in Fig. 3a. Within a shorter time window (1.2 s), we observed a detailed process of segmentation (Fig. 3b). Due to simultaneous transient contractions in the left and right of the orange circled intestine regions ($t = 0$ s), contrast agents are pushed to move towards each other. Then, a contraction appears in between the original contractions (Fig. 3b, $t = 0.67$ s). It segments the contrast agent and completes one cycle of segmentation.”

Fig. 3 Detailed movement of the intestine revealed by 2D-TIP, showing typical pacemaker-driven peristalsis and segmentation. (a) Representative 2D-TIP frames within a 120 s time window demonstrating the propagation of contrast agent inside the intestine, driven by a peristaltic motor pattern. (b) 2D-TIP frames showing a typical intestinal segmentation motor pattern.

Minor

Point 7: There are errors in spelling and grammar throughout.

Authors' response: We also appreciate the reviewer for calling our attention to the typos. We have performed a scrutinized spell check and corrected them.

Point 8: The statement “better extract intestines located far away from the abdomen” does not make sense.

Authors’ response: We thank the reviewer for pointing out this. We have now revised the sentence into ‘3D-TIP allows us to better extract intestines located far away from the abdomen wall’.

Reference:

- 1 Chitgupi, U. *et al.* Surfactant-Stripped Micelles for NIR-II Photoacoustic Imaging through 12 cm of Breast Tissue and Whole Human Breasts. *Adv. Mater.* **31**, 1902279 (2019).
- 2 Der-Silaphet, T., Malysz, J., Hagel, S., Arsenault, A. L. & Huizinga, J. D. Interstitial cells of Cajal direct normal propulsive contractile activity in the mouse small intestine. *Gastroenterology* **114**, 724-736 (1998).
- 3 Huizinga, J. D. *et al.* W/kit gene required for interstitial cells of Cajal and for intestinal pacemaker activity. *Nature* **373**, 347-349 (1995).
- 4 Thomson, L. *et al.* Interstitial cells of Cajal generate a rhythmic pacemaker current. *Nat. Med.* **4**, 848-851 (1998).
- 5 Spencer, N. J. & Hu, H. Enteric nervous system: sensory transduction, neural circuits and gastrointestinal motility. *Nature Reviews Gastroenterology & Hepatology*, 1-14 (2020).
- 6 Huizinga, J. D. *et al.* The origin of segmentation motor activity in the intestine. *Nat. Commun.* **5**, 1-11 (2014).
- 7 Huizinga, J. D. *et al.* Motor patterns of the small intestine explained by phase-amplitude coupling of two pacemaker activities: the critical importance of propagation velocity. *American Journal of Physiology-Cell Physiology* **309**, C403-C414 (2015).
- 8 Spencer, N. J., Dinning, P. G., Brookes, S. J. & Costa, M. Insights into the mechanisms underlying colonic motor patterns. *The Journal of physiology* **594**, 4099-4116 (2016).
- 9 Cannon, W. B. The movements of the intestines studied by means of the Rontgen rays. *American Journal of Physiology-Legacy Content* **6**, 251-277 (1902).
- 10 Alvarez, W. C. Bayliss and Starling's law of the intestine or the myenteric reflex. *American Journal of Physiology-Legacy Content* **69**, 229-248 (1924).
- 11 Huizinga, J. D. & Lammers, W. J. Gut peristalsis is governed by a multitude of cooperating mechanisms. *American Journal of Physiology-Gastrointestinal and Liver Physiology* **296**, G1-G8 (2009).
- 12 Huizinga, J. D. & Parsons, S. P. Pacemaker network properties determine intestinal motor pattern behaviour. *Exp Physiol* **104**, 623-624 (2019).
- 13 Wang, D. *et al.* Ingestible roasted barley for contrast-enhanced photoacoustic imaging in animal and human subjects. *Biomaterials* **175**, 72-81 (2018).
- 14 Zhang, Y. *et al.* Surfactant-Stripped Frozen Pheophytin Micelles for Multimodal Gut Imaging. *Adv. Mater.* **28**, 8524-8530 (2016).
- 15 Payne, S. C., Furness, J. B. & Stebbing, M. J. Bioelectric neuromodulation for gastrointestinal disorders: effectiveness and mechanisms. *Nature Reviews Gastroenterology & Hepatology* **16**, 89-105 (2019).
- 16 Yuan, Y. *et al.* Associations between colonic motor patterns and autonomic nervous system activity assessed by high-resolution manometry and concurrent heart rate variability. *Front. Neurosci.* **13**, 1447 (2020).
- 17 Milkova, N., Parsons, S. P., Ratcliffe, E., Huizinga, J. D. & Chen, J.-H. On the nature of high-amplitude propagating pressure waves in the human colon. *American Journal of Physiology-Gastrointestinal and Liver Physiology* **318**, G646-G660 (2020).
- 18 Dinning, P. *et al.* High-resolution colonic motility recordings in vivo compared with ex vivo recordings after colectomy, in patients with slow transit constipation. *Neurogastroenterol. Motil.* **28**, 1824-1835 (2016).

- 19 Padmanabhan, P., Grosse, J., Asad, A. B. M. A., Radda, G. K. & Golay, X. Gastrointestinal transit measurements in mice with ^{99m}Tc-DTPA-labeled activated charcoal using NanoSPECT-CT. *EJNMMI research* **3**, 60 (2013).
- 20 Owen, S. F., Liu, M. H. & Kreitzer, A. C. Thermal constraints on in vivo optogenetic manipulations. *Nat. Neurosci.* **22**, 1061-1065 (2019).
- 21 Arias-Gil, G., Ohl, F. W., Takagaki, K. & Lippert, M. T. Measurement, modeling, and prediction of temperature rise due to optogenetic brain stimulation. *Neurophotonics* **3**, 045007 (2016).
- 22 Gabella, G. The number of neurons in the small intestine of mice, guinea-pigs and sheep. *Neuroscience* **22**, 737-752 (1987).
- 23 Hamano, N., Inada, T., Iwata, R., Asai, T. & Shingu, K. The α 2-adrenergic receptor antagonist yohimbine improves endotoxin-induced inhibition of gastrointestinal motility in mice. *Br. J. Anaesth.* **98**, 484-490 (2007).
- 24 Spencer, N. J. *et al.* Mechanisms underlying distension-evoked peristalsis in guinea pig distal colon: is there a role for enterochromaffin cells? *American Journal of Physiology-Gastrointestinal and Liver Physiology* **301**, G519-G527 (2011).

REVIEWERS' COMMENTS

Reviewer #1 (Remarks to the Author):

This revision is a substantial improvement. The authors have done a fine job convincing this reviewer the technique is not only new, but could address important questions in the future, with particular regards to brain-gut communication. The major plus of this study is that the intestine of mice can be imaged in free-moving unanesthetized animal studies. It is true that alternative *In vivo* techniques using X-ray or MRI are problematic because of the need for anesthesia or movement restraint. I am satisfied this study represents a significant step forward in intestinal science. The physiological data showing that phasic segmental contractions occur at a lower rate following anaesthesia is interesting and an important reminder that any technique using CNS anaesthesia can modify gastrointestinal behaviour.

Overall, this study was an extraordinarily complex undertaking. Having performed imaging of intestinal activities *in vitro* for >15 years, I understand the challenges. The author's demonstration that it is possible to now image GI-transit in live animals that are free-to-move and have the potential added complexity of the intestines moving about the abdomen during imaging is quite an accomplishment. The author's should be commended on this development.

I watched all movies carefully in the original submission and again here. They are necessary and appropriate. The writing of the manuscript is clear and coherent. The figures necessary and clear.

A major positive of the TIP technique is that it does not need the dark, as conventional fluorescence imaging does. In contrast, TIP relies on the strong trans-illuminated NIR light instead of the weak fluorescence emission. TIP is also capable of multicolor imaging, by using contrast agents that are spectrally separated in absorbance wavelength. With the matched illumination wavelength, separate contrast agent can be imaged. This would not be straightforward or even possible using other modalities such as X-ray or MR imaging. A major advantage of multicolor TIP is the accurate localization of contrast agents at different sections of intestine, thereby eliminating the problem of overlapping sections of the intestine. As an example, the author's performed dual-color TIP imaging (setup as described in the Methods). The authors sequentially gavaged two contrast agents with a 30 min interval. This was a novel experiment with potentially useful future scientific potential.

The author's have responded convincingly about my concern regarding heat-generated artifacts induced by excessive or prolonged light illumination. The author's have submitted new data after performing additional tests, by illuminating a piece of chicken breast tissue (standard test) for 2 h at a laser intensity of 14 mW/cm² - which was the same intensity used in the imaging experiment. They continuously monitored the surface temperature using a thermal camera (FLIR one). The author's provide strong data with the new supplementary figure 12. This improved the study substantially. It is true that the relative intensities of light used in this study are very different to the recent paper by Owen S et al. 2019, Nature Neuroscience.

Page 19, line 345:

The authors state "Decades of *in vitro* research on neurally driven motor patterns, overshadowed pacemaker driven activities, but here we show a definitive study on free-moving mice, that leaves no doubt about the critical importance of pacemaker driven motor patterns 29,30." This is an overstatement that is technically incorrect and needs correction. No one says that ICC are not important, that I remember. Mutant mice born without pacemaker ICC and slow waves in the mouse small intestine live and propulsion still occurs in adult mutant mice lacking ICC, so they can't be "critically important". This needs to be acknowledged and will not in any way denigrate the nice data presented. The way it is currently written is unacceptable. The author's data convincingly demonstrates ICC-MY driven slow waves likely underlie the major motor pattern they record *in vivo*, defined as "segmentation" and "peristalsis". Mutant mice live without ICC in the small bowel and propulsive neurogenic motor patterns still exist in this region of gut without ICC. Indeed, the great work from Dr. Huizinga has shown that mutant W/W^v mice live with ICC-MY in the small intestine. This needs to be quoted and an additional sentence inserted to provide some balance. A statement and reference needs to be included at the end of this paragraph. Reference to

<https://pubmed.ncbi.nlm.nih.gov/14514874/>

is necessary here. My suggestion is at the end of the paragraph: "It is important to acknowledge that in mutant mice lacking pacemaker cells (ICC) and electrical slow waves in the small intestine, neurogenic propagating contractions still occur. These neurogenic contractions must be able capable of propelling content, as these mutant animals are not lethal and live without major intestinal complications (Spencer et al. 2003; ref below). It will be exciting to apply the new technique developed here to mutant mice with selective deletions in classes of enteric neurons or ICC or glia, to determine how intestinal transit is disrupted."

Reference to be inserted.

<https://pubmed.ncbi.nlm.nih.gov/14514874/>

Figure 4a has a spelling mistake in the figure. "Turinning" should be "Turning"

Reviewer #2 (Remarks to the Author):

I thank the authors for the extensive modifications in response to other reviewers and my comments. I especially appreciate the new experiments to clarify the approach's resolution limits and experiments tracking gut motor activity over an extended period (though still relatively short). However, in the quest to bring in physiologic meaning to this mostly technical advance, the authors ended up overinterpreting their data and ended up with incorrect conclusions. The issue is that since the mice fast for 24 hours and the contrast agent used in this study is not nutritive, the only patterns that the authors are recording are migrating motor complexes. Examples below.

In the abstract, the claim that these data provide "evidence for the hypothesis that a slow moving motor pattern is underlying the generation of segmentation." The data do not support this conclusion because the authors do not clearly know that the patterns are indeed segmenting.

Abstract also states "we show here that motor patterns controlled by intestinal pacemaker cells dominate activity evoked by distention due to the contrast fluid." However, this study did not access the roles of pacemaker cells nor neurons, and therefore this claim should be removed.

Finally, the abstract also states, "we also show the effects of anesthesia on motor patterns, highlighting the role of the extrinsic nervous system in controlling motor patterns." Anesthesia impacts both enteric and extrinsic neurons, so this claim should also be removed.

A part of the revised discussion is talking about peristalsis vs segmentation, and neuronal vs myogenic mechanisms. This is mostly speculation since the myogenic vs neurogenic pathways were not explored in this study.

02/09/2021

Responses to the comments on the manuscript “Trans-illumination Intestine Projection imaging of intestinal motility in mice” (Manuscript ID: NCOMMS-20-04089A)

We would like to thank you for the insightful comments, criticisms, and suggestions on our manuscript. The reviewers have raised many excellent questions, and we have endeavored to address each issue in detail in this response and in the revised manuscript. In our response, the reviewers’ comments are in **blue font**, and are followed by our detailed responses in black font. Texts which have been newly added or altered in the revised manuscript are in **red font**. These edits match the color of revised text in the completely revised manuscript.

Our completely revised manuscript have more accurately interpreted our data and removed exaggerated statements. We thank the editors and reviewers again for your time and effort on reading and commenting on our manuscript.

Best Regards,

Jun Xia, Ph.D.
Associate Professor
Department of Biomedical Engineering
University at Buffalo

REVIEWER COMMENTS

Reviewer #1 (Remarks to the Author):

Comment 1-1: This revision is a substantial improvement. The authors have done a fine job convincing this reviewer the technique is not only new, but could address important questions in the future, with particular regards to brain-gut communication. The major plus of this study is that the intestine of mice can be imaged in free-moving unanesthetized animal studies. It is true that alternative In vivo techniques using X-ray or MRI are problematic because of the need for anesthesia or movement restraint. I am satisfied this study represents a significant step forward in intestinal science. The physiological data showing that phasic segmental contractions occur at a lower rate following anaesthesia is interesting and an important reminder that any technique using CNS anaesthesia can modify gastrointestinal behaviour.

Authors' response: We thank this reviewer for his very encouraging positive comments!

Comment 1-2: Overall, this study was an extraordinarily complex undertaking. Having performed imaging of intestinal activities in vitro for >15 years, I understand the challenges. The author's demonstration that it is possible to now image GI-transit in live animals that are free-to-move and have the potential added complexity of the intestines moving about the abdomen during imaging is quite an accomplishment. The author's should be commended on this development.

Authors' response: We thank the reviewer for the encouraging words!

Comment 1-3: I watched all movies carefully in the original submission and again here. They are necessary and appropriate. The writing of the manuscript is clear and coherent. The figures necessary and clear.

Authors' response: We thank the reviewer for the encouraging words!

Comment 1-4: A major positive of the TIP technique is that it does not need the dark, as conventional fluorescence imaging does. In contrast, TIP relies on the strong trans-illuminated NIR light instead of the weak fluorescence emission. TIP is also capable of multicolor imaging, by using contrast agents that are spectrally separated in absorbance wavelength. With the matched illumination wavelength, separate contrast agent can be imaged. This would not be straightforward or even possible using other modalities such as X-ray or MR imaging. A major advantage of multicolor TIP is the accurate localization of contrast agents at different sections of intestine, thereby eliminating the problem of overlapping sections of the intestine. As an example, the author's performed dual-color TIP imaging (setup as described in the Methods). The authors sequentially gavaged two contrast agents with a 30 min interval. This was a novel experiment with potentially useful future scientific potential.

Authors' response: We thank the reviewer for the encouraging words!

Comment 1-5: The author's have responded convincingly about my concern regarding heat-generated artifacts induced by excessive or prolonged light illumination. The author's have submitted new data after performing additional tests, by illuminating a piece of chicken breast tissue (standard test) for 2 h at a laser intensity of 14 mW/cm² - which was the same intensity used in the imaging experiment. They continuously monitored the surface temperature using a thermal camera (FLIR one). The author's provide strong data with the new supplementary figure 12. This improved the study substantially. It is true that the relative intensities of light used in this study are very different to the recent paper by Owen S et al. 2019, Nature Neuroscience.

Authors' response: We thank the reviewer for the encouraging words!

Comment 1-6: Page 19, line 345:

The authors state "Decades of in vitro research on neurally driven motor patterns, overshadowed pacemaker driven activities, but here we show a definitive study on free-moving mice, that leaves no doubt about the critical importance of pacemaker driven motor patterns 29,30." This is an overstatement that is technically incorrect and needs correction. No one says that ICC are not important, that I

remember. Mutant mice born without pacemaker ICC and slow waves in the mouse small intestine live and propulsion still occurs in adult mutant mice lacking ICC, so they can't be "critically important". This needs to be acknowledged and will not in any way denigrate the nice data presented. The way it is currently written is unacceptable. The author's data convincingly demonstrates ICC-MY driven slow waves likely underlie the major motor pattern they record in vivo, defined as "segmentation" and "peristalsis". Mutant mice live without ICC in the small bowel and propulsive neurogenic motor patterns still exist in this region of gut without ICC. Indeed, the great work from Dr. Huizinga has shown that mutant W/W^v mice live with ICC-MY in the small intestine. This needs to be quoted and an additional sentence inserted to provide some balance. A statement and reference needs to be included at the end of this paragraph. Reference to <https://pubmed.ncbi.nlm.nih.gov/14514874/> is necessary here. My suggestion is at the end of the paragraph: "It is important to acknowledge that in mutant mice lacking pacemaker cells (ICC) and electrical slow waves in the small intestine, neurogenic propagating contractions still occur. These neurogenic contractions must be able capable of propelling content, as these mutant animals are not lethal and live without major intestinal complications (Spencer et al. 2003; ref below). It will be exciting to apply the new technique developed here to mutant mice with selective deletions in classes of enteric neurons or ICC or glia, to determine how intestinal transit is disrupted."

Reference to be inserted. <https://pubmed.ncbi.nlm.nih.gov/14514874/>

Authors' response: We agree that all the explanations of the reviewer related to transit in the mouse intestine are correct. As the reviewer indicates, we have done experiments in the past that are consistent with the view of this reviewer. We did not intend to make exaggerated statements and we are happy to entirely rewrite this section to address all the points made by the reviewer, see the revision below that appears in the discussion in context. The suggested reference is cited.

Corresponding changes in the manuscript

... This may have been in part because low resolution techniques do not allow fine detail needed to discern patterns such as the Cannon-type segmentation motor pattern [1]. The present study was

executed with a non-nutritious meal that will result in a cyclic fasting motor pattern, also termed the migrating motor complex [2, 3]. This consists of phase I, which is a quiet phase, and phase II with an “erratic” or fed-state-like activity, followed by phase III that is propulsive in nature. The segmentation motor pattern observed in the present study is likely part of phase II. It can be readily observed in the excised intestine, ex vivo [4]. Strong rhythmic propulsive activity is also observed in the mouse intestine ex vivo, where it has been called migrating motor complex [3] or minute rhythm [5, 6]; it is a neurogenic motor activity that can be observed in mutant mice that do not have ICC-MP [3, 6]. The minute rhythm occurs both in fasting rats as part of phase II and in fed rats [7]. It will be exciting to apply the technique developed here to mutant mice with selective deletions in classes of enteric neurons [8], deletions in subtypes of ICC assessing primary and stimulus-dependent pacemaker activities [9-11], or deletions in glia [12], to determine the role of the various cell types in control of intestinal transit and segmentation.

Comment 1-7: Figure 4a has a spelling mistake in the figure. “Turinning” should be “Turning”

Authors’ response: Sorry for the typo. We have corrected this in Figure 4a.

Reviewer #2 (Remarks to the Author):

Comment 2-1: I thank the authors for the extensive modifications in response to other reviewers and my comments. I especially appreciate the new experiments to clarify the approach's resolution limits and experiments tracking gut motor activity over an extended period (though still relatively short).

Authors' response: We thank the reviewer for the very positive comments!!

Comment 2-2: However, in the quest to bring in physiologic meaning to this mostly technical advance, the authors ended up overinterpreting their data and ended up with incorrect conclusions. The issue is that since the mice fast for 24 hours and the contrast agent used in this study is not nutritive, the only patterns that the authors are recording are migrating motor complexes.

Authors' response: The reviewer is indeed correct that the migrating motor complex is an important motor pattern in the fasting small intestine. The migrating motor complex consists of a period of non-activity, followed by a period of "seemingly random contractions" followed by the well-known phase III of the MMC which causes propulsion of content [13]. In the dog, phase II can last for hours [14]. It is often noted that phase II activity is similar to the fed motor pattern [14]. At the time that the MMC was extensively investigated, the low resolution of the recordings could not discern a Cannon type segmentation motor pattern. We have extensively studied this motor pattern in the small intestine of the mouse, *ex vivo*, with a non-nutritous physiological salt solution, published, as it happens, in *Nature Communications* [15]. Hence this motor pattern does not need nutrient activation. We have a similar experience in the human colon: in the prepared colon with all content flushed out, we observe all the motor patterns that are seen in the unprepared colon (possibly quantitatively different) [16-18]. Hence the segmentation motor pattern, is seen, we are convinced, in phase II of the MMC.

We have fully explained the role of the MMC in fasting in the discussion and in response to the other reviewer, please also see below.

Corresponding changes in the manuscript

... This may have been in part because low resolution techniques do not allow fine detail needed to discern patterns such as the Cannon-type segmentation motor pattern [1]. The present study was executed with a non-nutritious meal that will result in a cyclic fasting motor pattern, also termed the migrating motor complex [2, 3]. This consists of phase I, which is a quiet phase, and phase II with an “erratic” or fed-state-like activity, followed by phase III that is propulsive in nature. The segmentation motor pattern observed in the present study is likely part of phase II. It can be readily observed in the excised intestine, ex vivo [4]. Strong rhythmic propulsive activity is also observed in the mouse intestine ex vivo, where it has been called migrating motor complex [3] or minute rhythm [5, 6]; it is a neurogenic motor activity that can be observed in mutant mice that do not have ICC-MP [3, 6]. The minute rhythm occurs both in fasting rats as part of phase II and in fed rats [7]. It will be exciting to apply the technique developed here to mutant mice with selective deletions in classes of enteric neurons [8], deletions in subtypes of ICC assessing primary and stimulus-dependent pacemaker activities [9-11], or deletions in glia [12], to determine the role of the various cell types in control of intestinal transit and segmentation.

Comment 2-3: In the abstract, the claim that these data provide "evidence for the hypothesis that a slow moving motor pattern is underlying the generation of segmentation." The data do not support this conclusion because the authors do not clearly know that the patterns are indeed segmenting.

Authors' response: We fully agree that this sentence is not what we intended. Your comment makes it clear that we did not make ourselves clear. We have deleted this in the abstract, as you requested.

Comment 2-4: Abstract also states "we show here that motor patterns controlled by intestinal pacemaker cells dominate activity evoked by distention due to the contrast fluid." However, this study did not assess the roles of pacemaker cells nor neurons, and therefore this claim should be removed.

Authors' response: We agree that there is inference here. It is however, critical that we highlight the possibility to study pacemaker driven motor patterns. We have reworded this, to show that we did not study pacemaker activity in the present study.

Comment 2-5: Finally, the abstract also states, "we also show the effects of anesthesia on motor patterns, highlighting the role of the extrinsic nervous system in controlling motor patterns." Anesthesia impacts both enteric and extrinsic neurons, so this claim should also be removed.

Authors' response: We agree with the reviewer. Anesthesia is not selective for the autonomic nervous system hence we have corrected this. The important point we need to make is that we can now study in principle effects of the extrinsic nervous system that can only be done in vivo in unanesthetized animals. We have reworded accordingly.

Comment 2-6: A part of the revised discussion is talking about peristalsis vs segmentation, and neuronal vs myogenic mechanisms. This is mostly speculation since the myogenic vs neurogenic pathways were not explored in this study.

Authors' response: We have made sure that we refer to previous research for certain statements and not give the impression (which we did not intend) to refer to the present study.

Some of the authors have spent decades deciphering myogenic and neurogenic components of motor patterns. The objective of the discussion is in part to infer and speculate, but, we agree, this should be done in moderation, hence we have gone over this section to be sure. This was also done in response to the Comment 2-2 of Reviewer 2.

References:

1. W. B. Cannon, "The movements of the intestines studied by means of the Rontgen rays," *American Journal of Physiology-Legacy Content* **6**, 251-277 (1902).
2. J. H. Szurszewski, "A migrating electric complex of canine small intestine," *American Journal of Physiology-Legacy Content* **217**, 1757-1763 (1969).
3. N. J. Spencer, K. M. Sanders, and T. K. Smith, "Migrating motor complexes do not require electrical slow waves in the mouse small intestine," *The Journal of physiology* **553**, 881-893 (2003).
4. J. D. Huizinga, J.-H. Chen, Y. F. Zhu, A. Pawelka, R. J. McGinn, B. L. Bardakjian, S. P. Parsons, W. A. Kunze, R. Y. Wu, and P. Bercik, "The origin of segmentation motor activity in the intestine," *Nat. Commun.* **5**, 3326 (2014).
5. P. Fleckenstein, L. Bueno, J. Fioramonti, and Y. Ruckebusch, "Minute rhythm of electrical spike bursts of the small intestine in different species," *American Journal of Physiology-Gastrointestinal and Liver Physiology* **242**, G654-G659 (1982).
6. S. P. Parsons and J. D. Huizinga, "Nitric Oxide Is Essential for Generating the Minute Rhythm Contraction Pattern in the Small Intestine, Likely via ICC-DMP," *Front. Neurosci.* **14**(2020).
7. S. Diamant and R. Scott, "Migrating action potential complexes—a feature of normal jejunal myoelectric activity in the rat," *Can. J. Physiol. Pharmacol.* **65**, 2269-2273 (1987).

8. N. J. Spencer and H. Hu, "Enteric nervous system: sensory transduction, neural circuits and gastrointestinal motility," *Nature Reviews Gastroenterology & Hepatology* **17**, 338-351 (2020).
9. J. D. Huizinga, K. Ambrous, and T. Der-Silaphet, "Co-operation between neural and myogenic mechanisms in the control of distension-induced peristalsis in the mouse small intestine," *The Journal of physiology* **506**, 843-856 (1998).
10. J. D. Huizinga and W. J. Lammers, "Gut peristalsis is governed by a multitude of cooperating mechanisms," *American Journal of Physiology-Gastrointestinal and Liver Physiology* **296**, G1-G8 (2009).
11. J. D. Huizinga, S. P. Parsons, J.-H. Chen, A. Pawelka, M. Pistilli, C. Li, Y. Yu, P. Ye, Q. Liu, and M. Tong, "Motor patterns of the small intestine explained by phase-amplitude coupling of two pacemaker activities: the critical importance of propagation velocity," *American Journal of Physiology-Cell Physiology* **309**, C403-C414 (2015).
12. B. D. Gulbransen and K. A. Sharkey, "Novel functional roles for enteric glia in the gastrointestinal tract," *Nature reviews Gastroenterology & hepatology* **9**, 625 (2012).
13. N. W. Weisbrodt, "Patterns of intestinal motility," *Annu. Rev. Physiol.* **43**, 21-31 (1981).
14. I. M. Lang, S. K. Sarna, and R. E. Condon, "Gastrointestinal motor correlates of vomiting in the dog: quantification and characterization as an independent phenomenon," *Gastroenterology* **90**, 40-47 (1986).
15. J. D. Huizinga, J.-H. Chen, Y. F. Zhu, A. Pawelka, R. J. McGinn, B. L. Bardakjian, S. P. Parsons, W. A. Kunze, R. Y. Wu, and P. Bercik, "The origin of segmentation motor activity in the intestine," *Nat. Commun.* **5**, 1-11 (2014).
16. M. Pervez, E. Ratcliffe, S. P. Parsons, J. H. Chen, and J. D. Huizinga, "The cyclic motor patterns in the human colon," *Neurogastroenterol. Motil.* **32**, e13807 (2020).
17. N. Milkova, S. P. Parsons, E. Ratcliffe, J. D. Huizinga, and J.-H. Chen, "On the nature of high-amplitude propagating pressure waves in the human colon," *American Journal of Physiology-Gastrointestinal and Liver Physiology* **318**, G646-G660 (2020).
18. F. Chen, L. Zhang, J. Wu, F. Huo, X. Ren, J. Zheng, and D. Pei, "HCRP-1 regulates EGFR-AKT-BIM-mediated anoikis resistance and serves as a prognostic marker in human colon cancer," *Cell Death Dis.* **9**, 1-15 (2018).